# Temporal-Emerged Prompting for Segment Anything in Multiframe Infrared Small Target Detection

**Yinghui Xing** [1]   **Donghao Chu** [1]   **Shizhou Zhang** [1]   **Di Xu** [2]

## Abstract

Accurately localizing and segmenting small targets in low signal-to-noise ratio (SNR) infrared sequences remains a challenging task. Since targets are often indistinguishable from the background in individual frames, existing methods, even when equipped with advanced foundation model and powerful inter-frame association mechanisms, still fail to detect them. Motivated by the observation that targets tend to emerge gradually from the background over time and become distinguishable, we propose Temporal-Emerged Prompting for Segment Anything Model (TEP-SAM), a principled framework designed to explicitly exploit such temporal-emerged cues to modulate and prompt SAM. TEP-SAM operates by jointly modeling global motion patterns and local motion deviations to locate potential targets. It further enhances target region features by leveraging motion discrepancy, thereby generating temporal-emerged cues for SAM and enabling non-interactive segmentation. By bridging large-scale semantic pretraining with task-specific temporal modeling, TEP-SAM effectively adapts SAM to the challenging multiframe infrared small target detection task. Extensive experiments demonstrate the effectiveness of our approach, particularly under severely low-SNR conditions and in complex dynamic background. Code is available at https://github.com/cdh8285/TEP-SAM .

## 1. Introduction

Multiframe infrared small target detection (M-IRSTD) plays a vital role in many applications (Zhao et al., 2022; Kou

[1]School of Computer Science, Northwestern Polytechnical University, Xi'an, China [2]Huawei Technologies Ltd. Correspondence to: Shizhou Zhang <szzhang@nwpu.edu.cn>.

*Proceedings of the 43rd International Conference on Machine Learning*, Seoul, South Korea. PMLR 306, 2026. Copyright 2026 by the author(s).

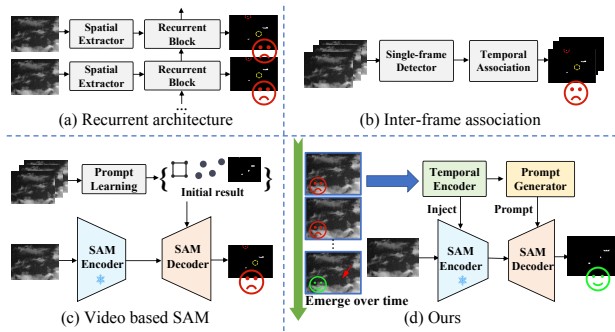

*Figure 1.* Comparison of different M-IRSTD paradigms. (a) Recurrent architecture framework, e.g., (Ying et al., 2025). (b) Single-frame detection combined with inter-frame association, e.g., (Li et al., 2023b). (c) Extensions of video-based SAM, e.g., (Hui et al., 2024). (d) Our TEP-SAM.

et al., 2023; Hua & Shao, 2017). Since targets of interest (e.g., small aircraft, missiles, vehicles, or ships) occupy only a few pixels in the field of view (Zhang et al., 2024a), it is often necessary to accurately segment the target boundaries so as to enable the subsequent extraction of precise target centroids (Zhang et al., 2022). However, in complex scenarios such as satellite or UAV imagery, targets are small and easily submerged in cluttered backgrounds. Under these conditions, target–background separability cannot be reliably established, making temporal information crucial.

Existing M-IRSTD approaches generally fall into two categories, as illustrated in Figure 1(a) and (b). One line of work relies on model architectures specifically designed for sequential data to extract spatio-temporal features (Yan et al., 2023; Li et al., 2023b). Their performance is largely determined by capability of the feature extractor, which are often limited by the absence of large-scale pre-training, leading to suboptimal results. The other line of work follows a sequential detection paradigm, in which targets are first detected on individual frames using a single-frame IRSTD (S-IRSTD) algorithm or a segmentation foundation model, and temporal consistency is subsequently enforced across frames (Chen et al., 2024; Ying et al., 2025). It is challenging for these methods to maintain the accuracy and completeness of the masks.

Segmentation foundation models such as SAM and SAM2 (Kirillov et al., 2023; Ravi et al.), pretrained on

large-scale datasets, exhibit strong feature extraction and segmentation capabilities. Building on them, several approaches have been developed for sequential data (Hui et al., 2024; Mei et al., 2025; Cuttano et al., 2025). As shown in Figure 1(c), they typically rely on *explicit prompts*, such as user-provided prompts, randomly sampled points, or detection results from auxiliary detection models. Under low signal-to-noise ratio (SNR) infrared conditions, the targets are extremely weak, making reliable prompt acquisition difficult and, consequently, accurate localization and segmentation particularly challenging. In IRSTD, recent efforts have attempted to adapt SAM by introducing specially designed prompting mechanisms (Zhang et al., 2024b; Fu et al., 2025). However, these methods are primarily tailored for S-IRSTD. In low-SNR infrared sequences, small targets are often indistinguishable from background clutter in individual frames. As a result, a naive extension to M-IRSTD suffers from severe missed detections and false alarms due to the lack of explicit temporal modeling. To substantiate this observation, we first extended the single-frame SAM-SPL method (Fu et al., 2025) to the M-IRSTD setting and conducted dedicated experiments (detailed in Section 4.3). The results show that, even when equipped with powerful inter-frame association mechanisms, such extensions still struggle to reliably localize and segment targets. In infrared images with dim targets and cluttered backgrounds, *targets may gradually emerge from background dynamics over time and become distinguishable*, as illustrated in Figure 1(d). In such cases, accurate temporal modeling can provide critical cues for segmentation models, enabling the discovery of targets that would otherwise be missed.

In this paper, we propose Temporal-Emerged Prompting for SAM (TEP-SAM) in M-IRSTD. We design a Discrepancy-Enhanced Temporal Encoder (DETE) to explicitly model global background dynamics and local target motion deviations, and to enhance target discriminability through discrepancy modeling. The resulting temporally emerged features are injected into the spatial representations produced by the frozen SAM encoder, enabling effective spatio-temporal feature enhancement. To eliminate the need for interactive prompting, we design a Temporal Prompt Generator (TP-Gen) that automatically derives temporal prompts from these temporally emerged features, facilitating accurate segmentation of weak targets. By feeding only a single frame into SAM while explicitly injecting temporal emerged cues into its spatial representations, TEP-SAM achieves effective spatio-temporal modeling without suffering from the typical trade-off in multiframe methods, where SNR enhancement often comes at the expense of increased background noise and reduced target discriminability.

The main contributions include: **1)** We propose TEP-SAM, a framework that equips SAM with the ability to segment weak and small targets in infrared sequences by ex-

ploiting temporally emerged features, without requiring user-provided prompts. **2)** We introduce a Discrepancy-Enhanced Temporal Encoder to extract temporally emerged features for spatial feature modulation and automatic temporal prompt generation. **3)** Extensive experiments show the effectiveness of proposed framework, particularly under challenging low-SNR and complex background scenarios.

## 2. Related Work

**Deep Learning Based Multiframe IRSTD Methods.** Recent advances have predominantly focused on spatio-temporal feature extraction (Yan et al., 2023; Chen et al., 2024; Ying et al., 2025; Zhu et al., 2024; Li et al., 2023b; 2025; Huang et al., 2024). Some methods employ 3D CNN to jointly model spatial and temporal features. For example, Yan et al. (2023) designed a temporal multiscale CNN extractor with parallel branches to form spatio-temporal features. Li et al. (2023b) introduced a direction-coded convolutions to explicitly encode target motion across frames. Li et al. (2025) proposed DeepPro, which probes per-pixel temporal profiles and captures long-range temporal correlations. Other approaches integrate architectures tailed for sequential data, such as RNN, LSTM, and GRU, into spatial feature extractors to realize motion feature modeling (Ying et al., 2025; Chen et al., 2024). Nevertheless, most existing frameworks rely on task-specific architectures and primarily emphasize motion contrast, with limited exploitation of rich spatio-temporal semantic priors. As a result, the performance remains constrained in challenging scenarios involving low SNR and dynamic backgrounds.

**Segment Anything Model on Sequential Data.** SAM (Kirillov et al., 2023) has strong zero-shot image segmentation capability when provided with visual prompts, such as points, boxes or masks. To extend it to sequential data, a number of methods have been proposed. More recently, SAM has been extended to a video-supported version, namely SAM2 (Ravi et al.), by introducing a streaming memory mechanism, in which features of the current frame are conditioned on the historical frame features and user prompts via memory attention. Building upon its memory design, several works directly inherit SAM2 to preserve temporal consistency and further refine its memory representation for task-specific objectives (Chen et al., 2025; Cuttano et al., 2025; Wang et al., 2025; Ye et al., 2025; Videnovic et al., 2025). In M-IRSTD, temporal information is particularly critical due to the extremely low SNR, sub-pixel target scale, and highly cluttered backgrounds. Consequently, specialized architectural designs are required to effectively adapt SAM to this challenging task.

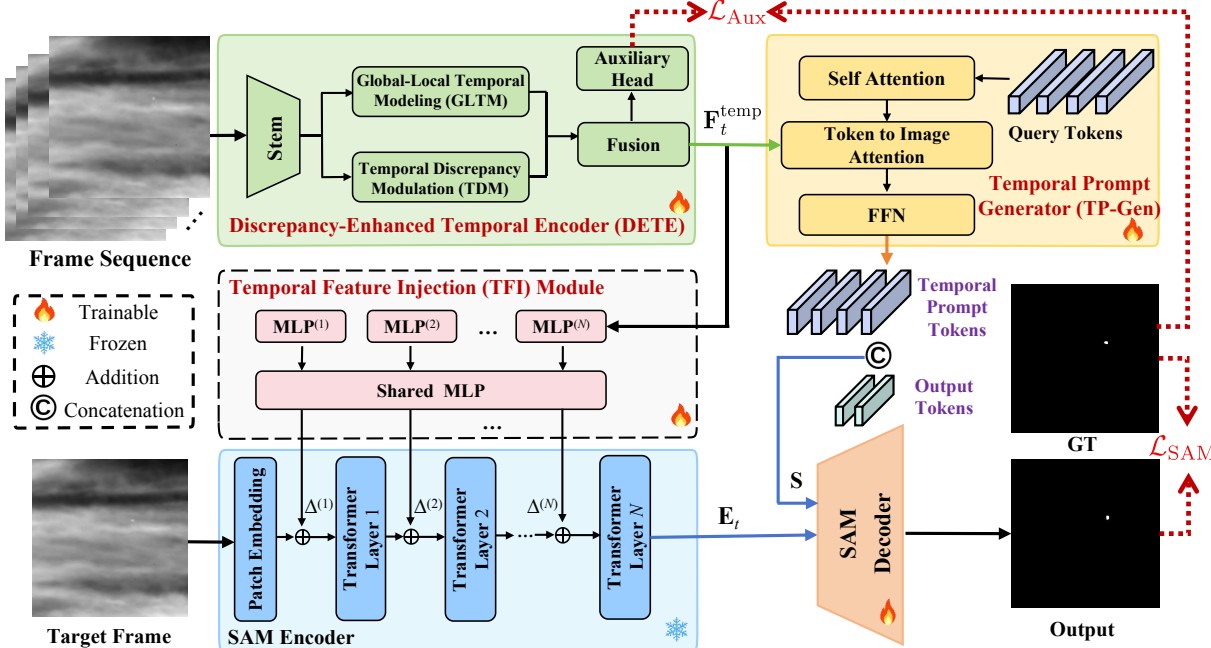

*Figure 2.* Overview of proposed TEP-SAM, which consists of a Discrepancy-Enhanced Temporal Encoder (DETE), a Temporal Feature Injection (TFI) module and a Temporal Prompt Generator (TP-Gen) for the modeling of temporal-emerged features, the modulation of SAM encoder features, and the generation of temporal prompts for SAM decoder, respectively.

## 3. Methodology

Given a sequential of infrared images $\mathcal{I} = \{\mathbf{I}_1, \ldots, \mathbf{I}_J\}$, the aim of M-IRSTD is to obtain a sequential of dense masks $\mathcal{M} = \{\mathbf{M}_1, \ldots, \mathbf{M}_J\}$, with each $\mathbf{M}_j \in \{0,1\}^{H \times W}, j = 1, \cdots, J$. In this paper, we propose a temporal-emerged prompting for SAM (TEP-SAM) in M-IRSTD.

### 3.1. Overview

The overall framework is illustrated in Figure 2. Following the main architecture of SAM, a single-frame image $\mathbf{I}_t, (t = 1, \cdots, J)$, referred to as the target frame, is fed into the SAM encoder and decoder to produce a prediction mask $\hat{\mathbf{M}}_t$. Infrared small targets are often dim and embedded in cluttered backgrounds, while possible cues may emerge along the temporal dimension, which can be used to complement spatial features. To this end, we introduce an auxiliary Discrepancy-Enhanced Temporal Encoder (DETE) together with a Temporal feature Injection (TFI) module to extract and incorporate temporal information into SAM. Specifically, DETE takes as input a frame sequence $\mathcal{I}_t = \{\mathbf{I}_{t-T}, \ldots, \mathbf{I}_t, \ldots, \mathbf{I}_{t+T}\}$ centered at $\mathbf{I}_t$, and extracts temporally emerged features associated with the target frame. These temporal features are then injected into the frozen SAM encoder via the TFI module, yielding enhanced spatio-temporal representations. Furthermore, as temporal emerged features can provide critical cues for revealing dim infrared small targets, we propose a Temporal Prompt Generator (TP-Gen) that takes the temporal features extracted

from DETE as input and generates temporal prompts. These prompts are combined with the output tokens of SAM for prompting the mask decoder. Finally, the temporal enhanced embeddings are processed by the SAM decoder to produce the final prediction mask for the target frame.

### 3.2. Discrepancy-Enhanced Temporal Encoder

In low-SNR infrared sequences, small targets are indistinguishable from the background in a single frame. However, their motion is inconsistent with the background dynamics. Over time, this motion discrepancy makes the targets separate from the background clutter. We design a Discrepancy-Enhanced Temporal Encoder (DETE) to extract the temporal cues. The details of DETE are provided in Figure 3, which consists of a Global-Local Temporal Modeling (GLTM) branch to model the discrepancy of motion patterns and a Temporal Discrepancy Modulation (TDM) branch to enhance the regions of target frame. Specifically, the frame sequence $\mathcal{I}_t$ are input to a stem block to extract a sequential spatio–temporal features $\mathcal{F}_t = \{\mathbf{F}_k, k \in [t - T, t + T]\}$:

$$\mathbf{F}_k = \text{Stem}(\mathbf{I}_k) \in \mathbb{R}^{C_0 \times H_0 \times W_0}, \quad (1)$$

where $\text{Stem}(\cdot)$ is composed of a series of temporal and spatial difference convolutions (Li et al., 2025). These features are then enhanced by the following two branches.

**Global-Local Temporal Modeling.** To effectively disentangle target motion from dynamic background interference, we explicitly model temporal discrepancy from both global

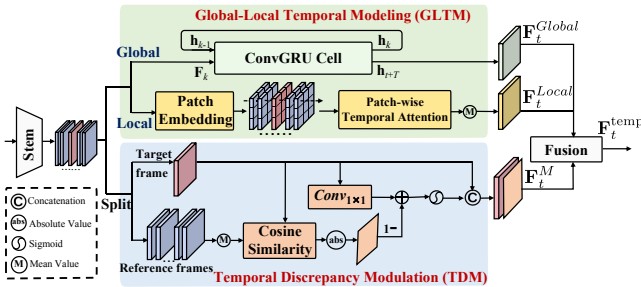

*Figure 3.* Structure of Discrepancy-Enhanced Temporal Encoder (DETE), where Global-Local Temporal Modeling branch models the deviations of targets from background dynamics and Temporal Discrepancy Modulation branch enhances target discrimination from background.

and local levels. For the global temporal feature modeling, a ConvGRU is used to accumulate global motion on each feature $\mathbf{F}_k, k \in [t - T, t + T]$:

$$
\begin{aligned}
\mathbf{z}_k &= \sigma\big(\mathbf{W}_z * [\mathbf{F}_k, \mathbf{h}_{k-1}]\big), \\
\mathbf{r}_k &= \sigma\big(\mathbf{W}_r * [\mathbf{F}_k, \mathbf{h}_{k-1}]\big), \\
\tilde{\mathbf{h}}_k &= \tanh\big(\mathbf{W}_h * [\mathbf{F}_k, \mathbf{r}_k \odot \mathbf{h}_{k-1}]\big), \\
\mathbf{h}_k &= (1 - \mathbf{z}_k) \odot \mathbf{h}_{k-1} + \mathbf{z}_k \odot \tilde{\mathbf{h}}_k,
\end{aligned}
\tag{2}
$$

where $\odot$ and $[\cdot, \cdot]$ represent the element-wise product and the channel-wise concatenation. $\sigma(\cdot)$ and $\tanh(\cdot)$ denote the sigmoid and hyperbolic tangent function, respectively. We then obtain the global temporal features from final state:

$$
\mathbf{F}_t^{Global} = Conv_{1 \times 1}(\mathbf{h}_{t+T}).
\tag{3}
$$

The global temporal features global motion patterns shared across frames, which primarily reflect background dynamics and global temporal consistency.

Meanwhile, the local subbranch focuses on fine-grained, region-specific temporal variations. Each feature $\mathbf{F}_k \in \mathbb{R}^{C_0 \times H_0 \times W_0}$ is partitioned into non-overlapping $p \times p$ patches. After projection and average pooling, we obtain the sequential of patch tokens $\mathbf{U}_k \in \mathbb{R}^{C_1 \times H_0/p \times W_0/p}$. Let $\mathbf{u}_k^{(s)} \in \mathbb{R}^{C_1}$ denote the vector of $\mathbf{U}_k$ in the spatial position $s$, a temporal patch-wise temporal attention $\text{TempAttn}(\cdot)$ is performed along the time window $[t - T, t + T]$, i.e.,

$$
\mathbf{A}^{(s)} = \text{TempAttn}\Big( [\mathbf{u}_{t-T}^{(s)}, \dots, \mathbf{u}_{t+T}^{(s)}] \Big) \in \mathbb{R}^{(2T+1) \times C_1}.
\tag{4}
$$

We then compute the mean value on it to form a local temporal feature descriptor,

$$
\bar{\mathbf{a}}^{(s)} = \frac{1}{2T+1} \sum_{\tau=1}^{2T+1} \mathbf{A}^{(s)}(\tau, :) \in \mathbb{R}^{C_1}.
\tag{5}
$$

Finally, $\{\bar{\mathbf{a}}^{(s)}\}_{s=1}^{H_0 W_0 / p^2}$ are rearranged and upsampled to recover the dense feature map:

$$
\mathbf{F}_t^{Local} = \text{Up}_p\big(\text{Rearrange}(\{\bar{\mathbf{a}}^{(s)}\})\big),
\tag{6}
$$

where $\text{Up}_p(\cdot)$ represents the upsampling by a factor of $p$. Such local representations are particularly effective for characterizing small targets, whose subtle motion pattern often deviates from the global background dynamics.

**Temporal Discrepancy Modulation.** Since the extracted temporal features are ultimately injected into the target frame, the spatio-temporal characteristics of that frame must be explicitly modeled and enhanced using information from adjacent reference frames. For target frame feature $\mathbf{F}_t$, we compute the mean of its adjacent frames by $\bar{\mathbf{m}} = \frac{1}{2T} \sum_{i \neq t} \mathbf{F}_i$. The cosine similarity between the target frame feature and $\bar{\mathbf{m}}$ across channels is calculated by:

$$
\mathbf{C} = \frac{\sum_b \mathbf{F}_t^{(b)} \odot \bar{\mathbf{m}}^{(b)}}{\|\mathbf{F}_t\|_2 \|\bar{\mathbf{m}}\|_2},
\tag{7}
$$

where $b$ indicates channels, and $\| \cdot \|_2$ denotes the $\ell_2$ norm computed along the channel dimension. Based on the similarity map, we can compute the "dissimilarity" map, which implicitly represents the regions with motion discrepancy between target frame and adjacent reference frames. The features of target frame can be enhanced by this dissimilarity map:

$$
\mathbf{F}_t^M = \Big[ \mathbf{F}_t, \sigma\Big( Conv_{1 \times 1}(\mathbf{F}_t) + \big(1 - abs(\mathbf{C})\big) \Big) \Big],
\tag{8}
$$

where $\mathbf{1}$ denotes the all-ones matrix and $abs(\mathbf{C})$ calculates the absolute value of each element in $\mathbf{C}$.

The global and local temporal features, together with the modulated temporal features are concatenated and fused by a lightweight fusion convolution to obtain the temporal-enhanced features of target frame:

$$
\mathbf{F}_t^{\text{temp}} = Conv_{3 \times 3}\Big( [\mathbf{F}_t^{Global}, \mathbf{F}_t^{Local}, \mathbf{F}_t^M] \Big).
\tag{9}
$$

By jointly leveraging global background dynamics, local region-wise temporal variations, and an explicitly enhanced target frame representation guided by adjacent reference frames, DETE obtains complementary cues for robust target discrimination in complex and cluttered scenes.

### 3.3. Temporal Feature Injection

To preserve the pretrained semantics of SAM while effectively exploiting temporal cues, we keep the SAM encoder frozen and introduce a Temporal Feature Injection (TFI) module to inject temporal information into each transformer block of the encoder. Specifically, the temporal features $\mathbf{F}_t^{temp}$ extracted by DETE are processed by the TFI module, which consists of a *layer-specific MLP*, a *GELU* nonlinearity, and a *shared MLP*:

$$
\Delta^{(l)} = \text{MLP}_{\text{shared}}\Big( \text{GELU}\big(\text{MLP}^{(l)}(\mathbf{F}_t^{\text{temp}})\big)\Big).
\tag{10}
$$

The modulated temporal features are then injected into the token stream of the frozen SAM encoder:

$$\mathbf{F}_{\mathrm{SAM}}^{(l)} = \mathrm{TransFormer}_l(\mathbf{F}_{\mathrm{SAM}}^{(l-1)} + \Delta^{(l)}), \quad (11)$$

where $\mathrm{TransFormer}_l(\cdot), (l = 1, \cdots, N)$ denotes the $l$-th Transformer block of SAM encoder. The final output of the SAM encoder for the target frame is given by $\mathbf{E}_t = \mathbf{F}_{SAM}^{(N)}$.

### 3.4. Temporal Prompt Generator

Since temporal information often provides informative cues for detecting obscured infrared small targets, we propose a Temporal Prompt Generator (TP-Gen) to produce temporal prompts for the SAM decoder. As shown in Figure 2, this design not only supplies effective prompts to guide SAM, but also eliminates the need for interactive prompting.

TP-Gen consists of a self attention, a token to image attention and an FFN. It takes a set of learnable query tokens $\mathbf{Q}_0 \in \mathbb{R}^{N_q \times d}$ as queries, and the tokenized spatio-temporal features $\mathbf{F}_t^{\mathrm{temp}}$ (denoted as $\mathbf{T}$) as keys and values, to generate the temporal prompt tokens $\mathbf{P} \in \mathbb{R}^{N_q \times d}$:

$$\mathbf{P} = \mathrm{FFN}\Big(\mathbf{Q}_0 + \mathrm{TokenToImageAttn}\big(\mathrm{SA}(\mathbf{Q}_0), \mathbf{T}\big)\Big), \quad (12)$$

where $N_q$ denotes the number of temporal prompt tokens. $\mathrm{SA}(\cdot)$, $\mathrm{TokenToImageAttn}(\cdot, \cdot)$, and $\mathrm{FFN}(\cdot)$ represent the self attention, token-to-image cross-attention, and the feed-forward network, respectively.

Following the standard pipeline of SAM, the generated temporal prompt tokens are concatenated with the output tokens of SAM to form the final prompts:

$$\mathbf{S} = \big[\, \mathbf{O} \,;\, \mathbf{P} \,\big], \quad (13)$$

where $[\cdot; \cdot]$ denotes the concatenation along the token dimension and $\mathbf{O}$ represents the output tokens of SAM. Finally, SAM decoder takes the prompt tokens $\mathbf{S}$ together with the spatio-temporal image embedding $\mathbf{E}_t$ from the SAM encoder to produce the predicted mask for the target frame:

$$\hat{\mathbf{M}}_t = \mathcal{H}_{SAM}\big(\mathbf{S}, \mathbf{E}_t\big), \quad (14)$$

where $\mathcal{H}_{SAM}(\cdot, \cdot)$ denotes the mask decoder of SAM.

### 3.5. Loss Function

In our framework, the SAM encoder is kept frozen, while all other components are optimized using a combination of BCE loss (Kullback & Leibler, 1951) and soft Dice loss (Milletari et al., 2016). To provide explicit supervision for temporal feature learning and to stabilize the training process, we introduce an auxiliary prediction head in DETE to produce an auxiliary mask from the temporal features:

$$\hat{\mathbf{M}}_t^{\mathrm{aux}} = \mathcal{H}_{\mathrm{Aux}}\big(\mathbf{F}_t^{\mathrm{temp}}\big), \quad (15)$$

where $\mathcal{H}_{\mathrm{Aux}}(\cdot)$ denotes the auxiliary prediction head. The overall objective is defined as:

$$\mathcal{L} = \mathcal{L}_{\mathrm{SAM}} + \mathcal{L}_{\mathrm{Aux}}, \quad (16)$$

where $\mathcal{L}_{\mathrm{SAM}} = \mathcal{L}_{\mathrm{BCE}}(\hat{\mathbf{M}}, \mathbf{M}) + \mathcal{L}_{\mathrm{Dice}}(\hat{\mathbf{M}}, \mathbf{M})$, and $\mathcal{L}_{\mathrm{Aux}} = \mathcal{L}_{\mathrm{BCE}}(\hat{\mathbf{M}}^{\mathrm{aux}}, \mathbf{M}) + \mathcal{L}_{\mathrm{Dice}}(\hat{\mathbf{M}}^{\mathrm{aux}}, \mathbf{M})$.

## 4. Experiments

### 4.1. Experiment Setup

We evaluate the proposed method on two representative M-IRSTD benchmarks, NUDT-MIRSDT (Li et al., 2023b) and TSIRMT (Huang et al., 2024). Performance is assessed from both the pixel-level segmentation perspective, using IoU and nIoU, and the target-level detection perspective, using $Pd$ and $Fa$. More details are provided in the *Appendix A*.

### 4.2. Comparison with State-of-the-Arts

We compare TEP-SAM against a broad set of both single-frame and multiframe IRSTD approaches. In addition, to provide a fair and comprehensive comparison with SAM-based methods, we extend single-frame SAM-based methods to the multiframe setting and include them as additional baselines. We further compare our method with SAM2 and TSP-SAM (Hui et al., 2024), which natively support multiframe inputs.

**Quantitative Results.** Table 1 reports the quantitative comparison results. To further assess the robustness of different methods under low-SNR conditions, we additionally present results on the *Hard subset* of each benchmark, where the infrared sequences exhibit significantly lower SNRs. From Table 1, we can observe that single-frame methods perform inadequately on these M-IRSTD benchmarks. The presence of extremely low target-background contrast, tiny targets sizes, and dynamic background interference jointly degrades both localization performance (inferior $P_d$ and $F_a$) and segmentation accuracy (reduced IoU and nIoU). By exploiting temporal information, most multiframe approaches achieve noticeable performance improvements. Benefiting from the pretrained semantics of SAM, TEP-SAM surpasses the current state-of-the-art method by 2.92, and 8.45 IoU on the NUDT-MIRSDT and TSIRMT datasets, respectively. More importantly, on the two *Hard subset*, TEP-SAM achieves a promotion of 4.36 and 10.07, respectively, demonstrating its superior robustness under challenging low SNR conditions. This improvement can be attributed to the temporal-emerged feature injection and the temporal prompting mechanism, which effectively enhance target discriminability in the presence of moving background and severe noise.

**Qualitative Results.** Several representative examples are presented in Figure 4. It can be observed that in extremely low-SNR scenarios, existing multiframe methods tend to

*Table 1.* **Comparisons on NUDT-MIRSDT and TSIRMT datasets.** Both pixel-level indices (IoU and nIoU (%)) and target-level indices ($P_d$ (%) and $F_a$ ($10^{-6}$)) are reported. The first and second best results are marked in red and blue, respectively.

| Method | NUDT-MIRSDT (All) | | | | NUDT-MIRSDT (Hard subset) | | | | TSIRMT (All) | | | | TSIRMT (Hard subset) | | | |
|---|---|---|---|---|---|---|---|---|---|---|---|---|---|---|---|---|
| | IoU | nIoU | $P_d$ | $F_a\downarrow$ | IoU | nIoU | $P_d$ | $F_a\downarrow$ | IoU | nIoU | $P_d$ | $F_a\downarrow$ | IoU | nIoU | $P_d$ | $F_a\downarrow$ |
| **Singleframe** | | | | | | | | | | | | | | | | |
| WSLCM (Han et al., 2020) | 12.60 | 13.66 | 52.75 | 82.97 | 0.00 | 0.00 | 0.00 | 112.25 | 2.89 | 4.08 | 27.21 | 243.14 | 1.64 | 2.37 | 18.04 | 59.81 |
| RIPT (Dai & Wu, 2017) | 6.77 | 5.93 | 33.43 | 113.77 | 0.15 | 0.17 | 1.83 | 210.87 | 2.10 | 3.14 | 28.21 | 211.12 | 1.26 | 2.08 | 33.37 | 235.31 |
| ANLPT (Zhang et al., 2023) | 4.82 | 4.03 | 20.53 | 116.14 | 0.07 | 0.11 | 1.13 | 275.55 | 2.12 | 2.89 | 28.55 | 329.81 | 1.01 | 1.29 | 17.15 | 137.61 |
| UIUNet (Wu et al., 2022) | 62.72 | 45.76 | 75.65 | 279.70 | 31.20 | 10.92 | 36.48 | 506.57 | 42.48 | 43.77 | 58.79 | 221.69 | 35.64 | 36.25 | 46.09 | 206.46 |
| DNANet (Li et al., 2022) | 44.43 | 38.55 | 56.28 | 206.64 | 16.92 | 4.06 | 13.61 | 268.30 | 46.07 | 46.92 | 62.48 | 190.33 | 40.18 | 40.96 | 41.27 | 270.62 |
| HCFNet (Xu et al., 2024) | 51.63 | 52.49 | 62.17 | 70.30 | 27.53 | 12.58 | 24.57 | 156.22 | 48.53 | 49.63 | 59.55 | 192.40 | 39.57 | 40.44 | 48.95 | 229.80 |
| **Multi-frame** | | | | | | | | | | | | | | | | |
| NFTDGSTV (Liu et al., 2023) | 4.66 | 3.19 | 28.98 | 255.14 | 0.06 | 0.16 | 1.70 | 430.09 | 0.55 | 0.94 | 20.57 | 180.68 | 0.28 | 0.52 | 13.96 | 205.12 |
| SRSTT (Li et al., 2023a) | 46.26 | 51.71 | 91.21 | 42.83 | 26.58 | 32.40 | 71.46 | 84.40 | 2.73 | 2.94 | 39.15 | 362.53 | 2.06 | 2.18 | 26.86 | 397.04 |
| 4D-TR (Wu et al., 2023) | 32.77 | 24.04 | 89.99 | 223.45 | 23.75 | 11.82 | 86.77 | 381.64 | 7.27 | 7.41 | 52.84 | 66.34 | 4.11 | 4.17 | 48.00 | 52.15 |
| ResUNet+RFR (Ying et al., 2025) | 80.75 | 83.94 | 92.71 | 6.70 | 63.56 | 52.31 | 79.40 | 4.83 | 58.14 | 57.04 | 79.32 | 629.09 | 46.32 | 48.93 | 66.73 | 670.36 |
| DNANet+DTUM (Li et al., 2023b) | 83.35 | 85.73 | 95.32 | 3.61 | 63.33 | 51.34 | 84.69 | 7.66 | 57.87 | 59.29 | 74.51 | 184.20 | 45.80 | 46.45 | 62.22 | 211.01 |
| ResUNet+DTUM (Li et al., 2023b) | 81.67 | 84.36 | 98.38 | 7.72 | 61.82 | 51.83 | 94.71 | 16.65 | 54.59 | 54.16 | 75.36 | 469.43 | 42.35 | 45.57 | 62.83 | 513.53 |
| DeepPro (Li et al., 2025) | 83.04 | 83.71 | 98.26 | 2.32 | 67.29 | 54.92 | 95.27 | 3.76 | 43.03 | 42.03 | 70.05 | 486.98 | 25.42 | 28.16 | 52.49 | 571.55 |
| DeepPro-Plus (Li et al., 2025) | 83.17 | 85.85 | 99.65 | 2.63 | 63.04 | 55.67 | 99.05 | 6.24 | 57.70 | 57.09 | 82.38 | 282.71 | 44.70 | 47.75 | 72.70 | 328.09 |
| LMAFormer (Huang et al., 2024) | 64.93 | 64.56 | 93.46 | 1.25 | 58.64 | 45.51 | 78.64 | 2.15 | 65.89 | 65.63 | 86.10 | 185.78 | 55.56 | 59.07 | 79.05 | 272.39 |
| **SAM-based** | | | | | | | | | | | | | | | | |
| SAM+XMem (Point) | 37.16 | 0.16 | 47.31 | >999 | 19.24 | 0.03 | 9.07 | >999 | 41.82 | 3.06 | 61.03 | >999 | 19.64 | 1.41 | 44.08 | >999 |
| SAM+XMem (Box) | 43.88 | 45.49 | 46.56 | 4.31 | 35.07 | 2.40 | 4.19 | 7.38 | 50.81 | 53.04 | 67.05 | 84.53 | 34.90 | 42.94 | 52.59 | 89.61 |
| SAM+XMem++ (Point) | 37.04 | 0.16 | 47.25 | >999 | 19.18 | 0.03 | 8.88 | >999 | 41.20 | 3.07 | 60.56 | >999 | 19.95 | 1.42 | 43.62 | >999 |
| SAM+XMem++ (Box) | 43.75 | 45.42 | 46.56 | 5.00 | 35.08 | 2.41 | 4.91 | 7.62 | 49.94 | 52.57 | 66.52 | 84.28 | 34.67 | 42.88 | 52.25 | 89.83 |
| SAM2 (Ravi et al.) (Point) | 47.11 | 0.13 | 49.39 | >999 | 35.09 | 0.04 | 10.21 | >999 | 48.41 | 4.53 | 73.02 | >999 | 27.03 | 2.08 | 58.12 | >999 |
| SAM2 (Ravi et al.) (Box) | 54.99 | 56.47 | 61.31 | 0.63 | 42.85 | 23.36 | 24.20 | 1.06 | 59.39 | 58.89 | 81.61 | 196.39 | 47.06 | 49.83 | 72.22 | 337.29 |
| TSP-SAM (Hui et al., 2024) | 41.80 | 41.21 | 65.30 | 159.43 | 20.76 | 10.48 | 25.52 | 235.36 | – | – | – | – | – | – | – | – |
| SAMURAI (Box) (Yang et al., 2024) | 38.80 | 19.89 | 60.44 | 755.53 | 42.66 | 4.87 | 31.95 | 932.65 | 28.94 | 17.63 | 55.79 | >999 | 21.54 | 14.97 | 49.21 | >999 |
| SAM-I2V (point) (Mei et al., 2025) | 55.21 | 3.19 | 68.42 | >999 | 19.41 | 0.27 | 15.88 | >999 | 61.70 | 53.10 | 87.88 | >999 | 50.21 | 42.33 | 81.68 | >999 |
| SAM-I2V(box) (Mei et al., 2025) | 62.41 | 55.79 | 81.26 | 169.69 | 24.40 | 11.92 | 41.78 | 344.04 | 63.49 | 55.67 | 83.77 | 878.00 | 49.81 | 43.05 | 72.94 | >999 |
| SAM-SPL (Fu et al., 2025) | 47.47 | 38.66 | 56.85 | 207.03 | 22.04 | 5.77 | 14.18 | 198.59 | 49.36 | 46.53 | 66.96 | 852.16 | 33.28 | 36.20 | 50.86 | >999 |
| SAM-SPL+Temporal-Pooling | 33.73 | 26.77 | 53.15 | 214.87 | 20.58 | 2.31 | 9.26 | 223.31 | 34.91 | 36.70 | 58.54 | 383.42 | 20.21 | 26.08 | 39.12 | 495.29 |
| SAM-SPL+3DConv | 47.60 | 39.58 | 57.14 | 188.66 | 22.65 | 5.37 | 14.18 | 189.98 | 49.36 | 46.93 | 67.11 | 768.54 | 32.71 | 36.07 | 50.72 | 904.36 |
| SAM-SPL+XMem | 42.45 | 33.13 | 52.11 | 205.25 | 29.31 | 5.15 | 17.77 | 257.94 | 41.77 | 38.34 | 58.74 | >999 | 26.08 | 29.44 | 43.31 | >999 |
| SAM-SPL+XMem++ | 42.24 | 32.43 | 51.76 | 217.01 | 29.30 | 4.99 | 17.58 | 272.77 | 40.85 | 37.98 | 57.94 | >999 | 25.70 | 29.25 | 42.42 | >999 |
| SAM-SPL+DTUM | 76.28 | 78.49 | 96.24 | 5.68 | 63.42 | 53.50 | 91.68 | 12.62 | 61.17 | 59.78 | 83.24 | 392.28 | 50.55 | 51.51 | 76.78 | 666.03 |
| **Ours** | | | | | | | | | | | | | | | | |
| TEP-SAM (SAM-B) | 86.15 | 86.28 | 99.71 | 0.51 | 71.19 | 58.28 | 99.05 | 1.23 | 74.34 | 73.21 | 92.04 | 108.97 | 65.22 | 66.34 | 86.46 | 205.92 |
| TEP-SAM (SAM-L) | 86.27 | 86.54 | 99.65 | 0.42 | 71.65 | 58.95 | 98.87 | 0.91 | 73.43 | 72.42 | 89.29 | 77.37 | 63.95 | 65.22 | 82.68 | 158.44 |
| TEP-SAM (SAM2-B) | 77.30 | 77.31 | 99.77 | 0.72 | 67.97 | 55.58 | 99.24 | 1.78 | 72.64 | 71.68 | 91.42 | 104.26 | 63.44 | 64.33 | 84.84 | 216.12 |
| TEP-SAM (SAM2-L) | 77.18 | 77.28 | 99.83 | 1.04 | 67.57 | 55.91 | 99.43 | 2.45 | 74.06 | 73.79 | 93.18 | 59.29 | 65.63 | 67.89 | 88.19 | 112.20 |

miss weak targets or produce multiple false alarms. In relatively higher-SNR scenes, while competing methods can roughly localize the target, the masks they generate are often fragmented or incomplete. In contrast, TEP-SAM consistently produces more compact and coherent segmentation results that closely align with the groundtruths. These qualitative observations are consistent with the quantitative results and further indicate that the proposed temporal-guided feature injection and prompting mechanism enable SAM to better focus on truly moving small targets rather than being distracted by background motion.

### 4.3. Analysis on Temporal Prompting

**Comparisons with Eight Different Variants of SAM.** Table 1 and Figure 5 show the performance of various SAM-based variants under different prompting strategies and temporal modeling schemes. Since vanilla SAM is designed for single-image segmentation, we prompt only the first frame of each sequence using groundtruth points or bounding boxes, and then extend SAM to the multiframe setting by adopting two state-of-the-art video object segmentation methods, namely XMem (Cheng & Schwing, 2022) and XMem++ (Bekuzarov et al., 2023). In addition, SAM-SPL (Fu et al., 2025) is originally designed for S-IRSTD, we adapt it to the multiframe setting using five representative temporal modeling schemes, including simple temporal pooling, 3D convolution, XMem (Cheng & Schwing, 2022), XMem++ (Bekuzarov et al., 2023), and the specialized DTUM module (Li et al., 2023b) tailored for M-IRSTD. As shown in Table 1, naively extending SAM to multiframe setting leads to unsatisfactory performance, mainly due to insufficient exploitation of temporal information. Although

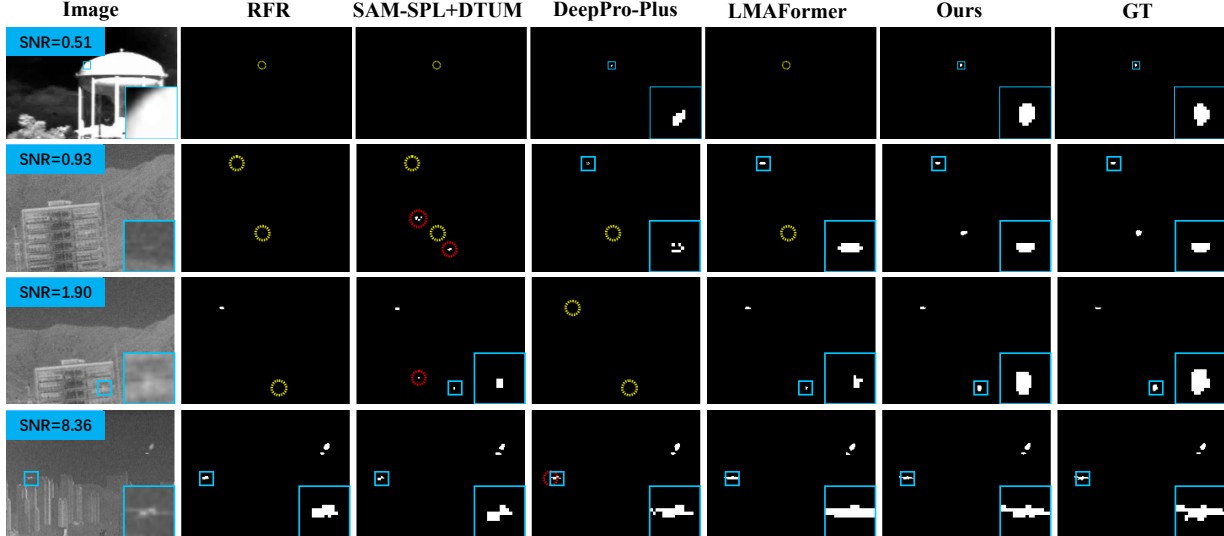

*Figure 4.* Qualitative comparisons across different SNR levels, where the red and yellow dashed circles denote the false alarms and the missed detections, respectively. One can zoom in for more details.

SAM2 introduces a memory mechanism for video segmentation, its design is not well aligned with the characteristics of IRSTD, where targets are small, low-contrast, and heavily obscured. Likewise, for the single-frame SAM-SPL method, straightforward multiframe extensions fail to effectively model temporal dependencies, even when combined with the task-specific DTUM. More importantly, it can be seen in Figure 5, TEP-SAM demonstrate superior performance no matter in target-level detection and pixel-level segmentation, especially under challenging low-SNR conditions. These results indicate that generic temporal aggregation or memory-based mechanisms are insufficient for M-IRSTD, where discriminative temporal cues are subtle and easily overwhelmed by complex background motion. In contrast, TEP-SAM explicitly mining temporal-emerged features, and use them to modulate and prompting SAM, which proves effective in M-IRSTD.

**Feature Visualizations.** To understand the benefits of temporal modeling in TEP-SAM, we visualize intermediate feature maps in Figure 6. The feature of vanilla SAM encoder exhibits strong responses over the background, indicating limited discrimination between targets and backgrounds. In contrast, the temporally modulated SAM encoder significantly enhances the responses around moving small targets. After incorporating temporal prompts, the target regions become more compactly activated, while background interference is largely suppressed. These observations demonstrate that temporal features provide effective cues for target localization, simultaneously enhancing target representations and mitigating cluttered background interferences.

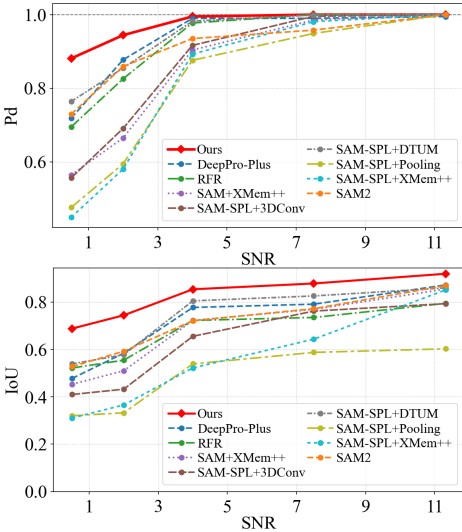

*Figure 5. Pd* and IoU comparisons across different SNR.

### 4.4. Ablation Study

We conduct ablation studies on the NUDT-MIRSDT dataset to examine the contributions of the main components of TEP-SAM. In addition, we analyze the design of the Discrepancy-Enhanced Temporal Encoder. All ablation experiments are performed using the SAM-B model.

**Effect of Discrepancy-Enhanced Temporal Encoder (DETE).** To evaluate the effectiveness of the DETE, we replace it with a per-frame convolutional encoder followed by temporal average pooling. As shown by the comparison between the first and last rows of Table 2, this modification results in a substantial performance degradation across all metrics, indicating that dedicated temporal modeling is critical for detecting weak and dim targets.

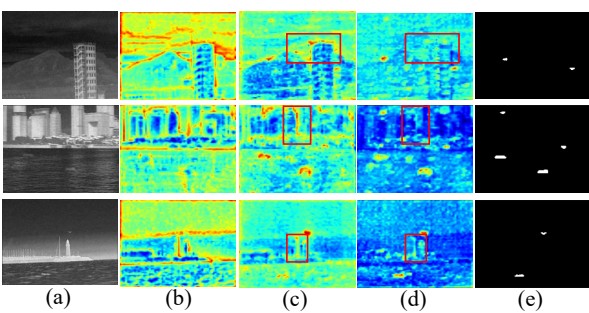

| (a) | (b) | (c) | (d) | (e) |

*Figure 6.* Visualization results: (a) infrared image; (b) output features of vanilla SAM encoder; (c) output features of temporally-modulated SAM encoder; (d) features for mask prediction in SAM decoder; and (e) final predicted mask.

*Table 2.* Ablations on main components.

| # | Configuration | IoU | nIoU | $P_d$ | $F_a \downarrow$ |
|---|---|---|---|---|---|
| 1 | w/o DETE | 64.84 | 70.86 | 96.12 | 46.00 |
| 2 | w/o TFI | 50.30 | 74.05 | 72.99 | 284.00 |
| 3 | w/o TPGen | 84.86 | 85.37 | **99.77** | 1.48 |
| 4 | w/o $\mathcal{L}_{Aux}$ | 82.69 | 83.96 | 98.55 | 5.63 |
| 5 | TEP-SAM | **86.15** | **86.28** | 99.71 | **0.51** |

**Effect of Temporal Feature Injection (TFI) Module.** To further assess the contribution of the TFI module, we remove it while keeping the rest of the architecture unchanged. As shown in the second row of Table 2, this leads to a significant performance degradation. The result indicates that explicitly coupling spatial and temporal features via the TFI module is crucial for effective M-IRSTD.

**Effect of Temporal Prompt Generator (TP-Gen).** We investigate the effect of the TP-Gen by removing the temporal prompt tokens. In this case, the model degenerates into a conventional multiframe detection network without prompt interaction. As shown in the third row of Table 2, this leads to a noticeable decrease in IoU accompanied by a slight increase in $P_d$, suggesting that temporal prompts primarily contribute to producing more accurate segmentation masks.

**Effect of Auxiliary Loss.** In this ablation study, we remove the auxiliary loss to evaluate its contribution. As shown in the fourth row of Table 2, the removal of the auxiliary loss leads to consistent degradation across all evaluation metrics. This performance drop indicates that the auxiliary loss plays a critical role in providing direct supervision to the DETE module, encouraging it to learn temporally discriminative representations that separate target-related dynamics from background motion. Without such explicit supervision, DETE fails to learn sufficiently discriminative representations, resulting in inferior detection performance.

In summary, these ablation results demonstrate that the accurate encoder-side temporal modulation and decoder-side temporal prompting are both indispensable and complemen-

*Table 3.* Ablations on discrepancy-enhanced temporal encoder.

| GLTM | | TDM | IoU | nIoU | $P_d$ | $F_a \downarrow$ |
|---|---|---|---|---|---|---|
| Local | Global | | | | | |
| | | | 80.30 | 82.26 | 98.08 | 11.50 |
| ✓ | | | 80.49 | 82.82 | 98.15 | 9.08 |
| | ✓ | | 85.31 | 85.48 | **99.77** | 1.61 |
| ✓ | ✓ | | 85.42 | 85.75 | **99.77** | 0.96 |
| | | ✓ | 79.76 | 81.60 | 98.61 | 13.54 |
| ✓ | | ✓ | 85.10 | 85.98 | **99.77** | 1.47 |
| | ✓ | ✓ | 83.75 | 84.45 | 99.42 | 5.29 |
| ✓ | ✓ | ✓ | **86.15** | **86.28** | 99.71 | **0.51** |

tary for fully exploiting temporal information.

**Design of Discrepancy-Enhanced Temporal Encoder.** DETE comprises a Global–Local Temporal Modeling (GLTM) branch and a Temporal Discrepancy Modulation (TDM) branch, which are designed to model global and local motion patterns and highlight target regions, respectively. To evaluate the effectiveness of each component, we conduct a series of ablation experiments. The results are provided in Table 3. From the table, we observe that introducing only global temporal features lead to substantial improvements across all evaluation metrics. Building upon this, further incorporating local temporal features yields additional performance gains. Notably, although employing the TDM branch alone results in performance degradation, its effectiveness is significantly enhanced when combined with either sub-branch of GLTM. We attribute this to the fact that TDM explicitly models the discrepancy between the target frame and adjacent reference frames, which may disrupt temporal consistency when used in isolation. The adverse effect can be effectively alleviated by incorporating global or local temporal modeling mechanisms. Overall, integrating all three components achieves the best performance, thereby validating their complementary roles.

## 5. Conclusion

In this paper, we presented Temporal-Emerged Prompting for SAM (TEP-SAM), a principled framework that integrates temporal-emerged cues into the SAM for M-IRSTD. By explicitly modeling global motion patterns and local motion deviations, TEP-SAM locates potential targets and enhances their features through motion discrepancy, thereby enabling non-interactive and temporally consistent segmentation across frames. The proposed temporal modeling mechanism effectively bridges large-scale semantic pretraining with the challenging dynamics of low-SNR infrared sequences, allowing SAM to adapt to low-signal and cluttered sequential scenarios. Extensive experimental results on multiple M-IRSTD benchmarks demonstrate that TEP-SAM consistently outperforms existing methods, achieving

accurate segmentation and robust localization of heavily obscured targets in complex backgrounds.

## Acknowledgements

This work was supported in part by the National Natural Science Foundation of China (NSFC) under Grant 62476223, 62576282.

## Impact Statement

This paper presents work whose goal is to advance the field of Machine Learning. There are many potential societal consequences of our work, none which we feel must be specifically highlighted here.

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

# A. More Details of Experiments

## A.1. Datasets

We evaluate our method on two representative multiframe IRSTD benchmarks, NUDT-MIRSDT (Li et al., 2023b) and TSIRMT (Huang et al., 2024). In all experiments, we follow the official data splits for fair comparison. Specifically, for NUDT-MIRSDT dataset, we use 80 sequences for training and 20 sequences for testing (Li et al., 2023b) (with an additional hard subset of 8 sequences), while for TSIRMT dataset, we use 140 sequences for training and 60 sequences for testing (Huang et al., 2024) (with an additional hard subset of 25 sequences).

**NUDT-MIRSDT** is constructed for temporal IRSTD and contains 100 sequences, each with 100 frames, with centroid points and pixel-level mask annotations. It covers diverse static backgrounds such as sky, cloud and ocean scenes and spot-like dim targets with weak energy and small spatial extent, the hard subset is particularly challenging due to its extremely weak targets (SNR $\leq 3$), where the target is barely distinguishable even for human experts from a single frame, and single-frame methods tend to fail without leveraging temporal salience cues.

**TSIRMT** is a large-scale simulated benchmark comprising 200 sequences, each with 100 frames, where realistic targets (spot-like targets, aircraft, UAVs, vehicles and ships) move over complex backgrounds including mountains, rivers, clouds, forests and cities. TSIRMT is specifically designed to model diverse target motion states and rich background motion, and provides pixel-level mask annotations, making it well suited for evaluating sequence-based methods under moving-background conditions. The TSIRMT hard subset focuses on dim-target conditions (TSNR $\leq 10$), where multiframe methods often suffer notable performance degradation. In this setting, targets may exhibit variations in scale, morphology, and energy over time, while pronounced background motion further introduces errors in inter-frame modeling, making robust separation of targets from dynamic backgrounds especially difficult.

## A.2. Evaluation Metrics

We adopt four metrics to evaluate both pixel-level segmentation and target-level detection performance, including intersection-over-union (IoU), normalized IoU (nIoU), probability of detection ($Pd$), and false alarm rate ($Fa$).

**Intersection-over-Union (IoU)** measures the overall overlap between predictions and ground truth:

$$\text{IoU} = \frac{\sum_{k=1}^{N_s} |M_k \cap G_k|}{\sum_{k=1}^{N_s} |M_k \cup G_k|}. \tag{17}$$

Here $M_k$ and $G_k$ denote the predicted and ground-truth binary masks of the $k$-th sample, $|\cdot|$ counts foreground pixels, and $N_s$ is the number of samples.

**Normalized IoU (nIoU)** is defined as the mean IoU over all samples:

$$\text{nIoU} = \frac{1}{N_s} \sum_{k=1}^{N_s} \frac{|M_k \cap G_k|}{|M_k \cup G_k|}. \tag{18}$$

**Probability of detection** ($Pd$). For target-level evaluation, we first extract instances from the binary masks and associate predicted and ground-truth targets according to the distance between their centroids. A predicted target is regarded as a true positive if the Euclidean distance between its centroid and that of a ground-truth target is less than 3 pixels; otherwise, unmatched predictions and unmatched ground truths are counted as false positives and false negatives, respectively. Under this matching rule, the probability of detection is defined as

$$Pd = \frac{N_{\text{TP}}}{N_{\text{TP}} + N_{\text{FN}}} = \frac{N_{\text{TP}}}{N_{\text{GT}}}, \tag{19}$$

where $N_{\text{TP}}$ and $N_{\text{FN}}$ are the numbers of true positives and false negatives, and $N_{\text{GT}}$ is the total number of ground-truth targets.

**False alarm ($Fa$) rate** measures the average number of false detections per frame:

$$Fa = \frac{N_{\text{FP}}}{N_{\text{frm}}}, \tag{20}$$

where $N_{\text{FP}}$ is the number of false positives (unmatched predictions), and $N_{\text{frm}}$ is the total number of frames in the test set.

## A.3. Implementation Details

In our experiments, infrared frames are first normalized and then resized to $1024 \times 1024$ to match the input resolution of SAM, and the predicted probability maps are subsequently rescaled back to the original image resolution. Since our temporal modulation and prompting design is independent of the main architecture of SAM, the proposed method is compatible with different versions of SAM. We have implemented our method on the ViT-B and ViT-L variant of both SAM and SAM2. The number of query tokens is $N_q = 4$, and the sequence length of the input of Discrepancy-Enhanced Temporal Encoder (DETE) is 7. For the leading and trailing frames, boundary frames are replicated to satisfy the required input sequence length. All results are reported using a fixed probability threshold of $0.5$ for binarizing the predicted masks. We optimize the network using the AdamW optimizer with an initial learning rate of $1 \times 10^{-4}$. The model is trained for 40 epochs with a batch size of 2.

## A.4. Comparision Methods

We compare our method with a wide range of single-frame and multiframe IRSTD approaches, including classical model-driven filtering methods (WSLCM (Han et al., 2020), RIPT (Dai & Wu, 2017), ANLPT (Zhang et al., 2023), NFTDGSTV (Liu et al., 2023), SRSTT (Li et al., 2023a), 4D-TR (Wu et al., 2023)), data-driven single-frame methods (UIUNet (Wu et al., 2022), DNANet (Li et al., 2022), HCFNet (Xu et al., 2024)), the SAM-based IRSTD framework SAM-SPL (Fu et al., 2025), and multiframe methods (DNA+DTUM, ResUNet+DTUM (Li et al., 2023b), RFR (Ying et al., 2025), DeepPro, DeepPro-Plus (Li et al., 2025), LMAFormer (Huang et al., 2024)).

# B. Additional Experiments

## B.1. Hyperparameter Sensitivity Analysis

In this section, we explore the optimal hyperparameters, including the number of prompt tokens and the length of temporal window. The experiments are performed on the SAM-B model.

**Number of Query Tokens.** The number of query tokens directly determines the length of the generated temporal tokens. To identify an appropriate configuration, we conduct experiments to analyze the impact of varying the number of query tokens. The results are reported in Table 4. As shown in the table, the overall performance is relatively insensitive to the token number. Increasing the number of tokens from 3 to 4 yields slight improvements in IoU and nIoU, accompanied by a reduction in the false alarm rate $F_a$. However, further increasing the token number results in only marginal IoU gains or even performance degradation, while noticeably increasing the false alarm rate and introducing additional model parameters. Considering the trade-off between performance, robustness, and model complexity, using four query tokens achieves the best overall balance. Therefore, we adopt four query tokens in all experiments.

*Table 4.* Ablation on the token number.

| Token Num | IoU | nIoU | $P_d$ | $F_a \downarrow$ |
|---|---|---|---|---|
| 3 | 85.87 | 85.99 | 99.71 | 0.83 |
| 4 | **86.15** | 86.28 | 99.71 | **0.51** |
| 5 | 85.39 | **86.35** | 99.71 | 0.93 |
| 6 | 85.75 | 85.71 | **99.83** | 1.02 |
| 7 | 84.03 | 85.77 | 99.71 | 3.48 |

**Temporal window length.** For each target frame, we construct a temporal clip of length $L = 2T + 1$. In this experiment, we explore different temporal window sizes by setting $T = 2, 3, 4, 5$ to identify the optimal clip length. As shown in Table 5, the model achieves the highest IoU and nIoU while maintaining a low false alarm rate $F_a$ when $T = 3$. In contrast, shorter temporal window ($T = 2$) fails to fully exploit temporal contextual information, whereas longer windows ($T = 4, 5$) introduce excessive background interference. This additional noise adversely affects both segmentation accuracy and the false alarm rate.

**Architecture of Global Temporal Modeling.** We compare ConvGRU with attention-based and Mamba-style modules for global temporal modeling. As shown in Table 6, ConvGRU achieves the best overall performance, especially with the

*Table 5.* Ablation on temporal window length, where the window length $L = 2T + 1$.

| $T$ | IoU | nIoU | $P_d$ | $F_a \downarrow$ |
|---|---|---|---|---|
| 2 | 85.74 | 85.00 | 95.95 | 0.16 |
| 3 | 86.15 | 86.28 | 99.71 | 0.51 |
| 4 | 85.32 | 85.59 | 99.83 | 2.16 |
| 5 | 82.55 | 83.91 | 99.71 | 6.20 |

lowest false alarm rate. This is because our global modeling mainly refers to cross-frame temporal accumulation rather than full-image spatial interaction, which better matches the spatial sparsity and temporal continuity of infrared small targets. In contrast, attention introduces more redundant background interactions, while the Mamba-style module does not show advantages under the relatively short temporal window. Therefore, we adopt ConvGRU in our final framework.

*Table 6.* Ablation on the architecture of global temporal modeling.

| Global Motion | IoU | nIoU | $P_d$ | $F_a \downarrow$ |
|---|---|---|---|---|
| Attention | 84.32 | 85.38 | **99.77** | 2.85 |
| Mamba Style | 84.24 | 85.06 | 98.73 | 2.00 |
| ConvGRU (Ours) | **86.15** | **86.28** | 99.71 | **0.51** |

## B.2. Computational Cost

Table 7 reports the trainable parameters, GFLOPs and FPS of TEP-SAM compared with representative IRSTD models. TEP-SAM incurs additional computation due to introduction of a vision foundation model, but still operates within a practical runtime range and offers a clear trade-off between efficiency and performance.

*Table 7.* Computational cost comparison.

| Method | Params(M) | GFLOPs | FPS |
|---|---|---|---|
| UIUNet (Wu et al., 2022) | 50.54 | 436.01 | 21.63 |
| DNANet (Li et al., 2022) | 4.70 | 114.26 | 12.88 |
| HCFNet (Xu et al., 2024) | 15.29 | 186.23 | 14.69 |
| SAM-SPL (Fu et al., 2025) | 12.25 | 30.47 | 32.54 |
| DeepPro-Plus (Li et al., 2025) | 0.28 | 20.5 | 185.22 |
| DNANet+DTUM (Li et al., 2023b) | 1.21 | 82.80 | 22.09 |
| LMAFormer (Huang et al., 2024) | 390.05 | 380.1 | 4.62 |
| Ours(SAM2-Tiny) | 5.88/33.10 | 301.1 | 30.23 |
| Ours(SAM1-Base) | 5.90/95.60 | 876.6 | 12.42 |

## B.3. More Visualizations

Figure 7 presents additional visualization results under varying Signal-to-Noise Ratio (SNR) levels.

In low-SNR cases, the target is heavily submerged in cluttered background, exhibiting extremely weak contrast and non-salient visual cues. For SAM-based approaches, relying on memory propagation or generic temporal feature extraction makes it difficult to consistently discover such weak targets, leading to frequent missed detections (yellow circles). In contrast, dedicated multiframe IRSTD methods, without the support of a general-purpose semantic extractor such as SAM, often only localize the approximate target position, while producing coarser and less accurate boundaries than ours in the zoomed-in regions. Moreover, due to dynamic background interference, their temporal extractors can be insufficiently stable, which easily introduces spurious responses and results in more false alarms (red circles).

When the SNR becomes higher, our predicted masks are almost shape-consistent with the GT, achieving the most precise boundary delineation.

Overall, our method delivers consistently accurate and fine-grained segmentation across varying SNR levels, achieving the highest IoU while preserving strong target-level detection performance.

## C. Limitations and Future Works

TEP-SAM is mainly applicable to short temporal windows with relatively coherent background motion, such as static observation, mild platform jitter, or slight camera motion. Under strong ego-motion, severe parallax, or highly complex scene dynamics, the target-background motion discrepancy may become less reliable, and additional motion compensation or scene-adaptive temporal modeling may be required.

The temporal-emerged features are established under the premise of target motion. Only under this condition can our proposed method model background motion, exploit the motion discrepancy between target and background, and extract temporal-emerged cues to enhance the capabilities of SAM. In scenarios where targets move extremely slowly or remain stationary, temporal features may provide limited discriminative information, requiring other strategies to enhance the discriminability between the target and the background. This remains a common challenging in the field of multiframe infrared small target detection. Furthermore, due to the introduction of a vision foundation model, the computational efficiency of our approach has room for improvement. These limitations point to directions for future research.

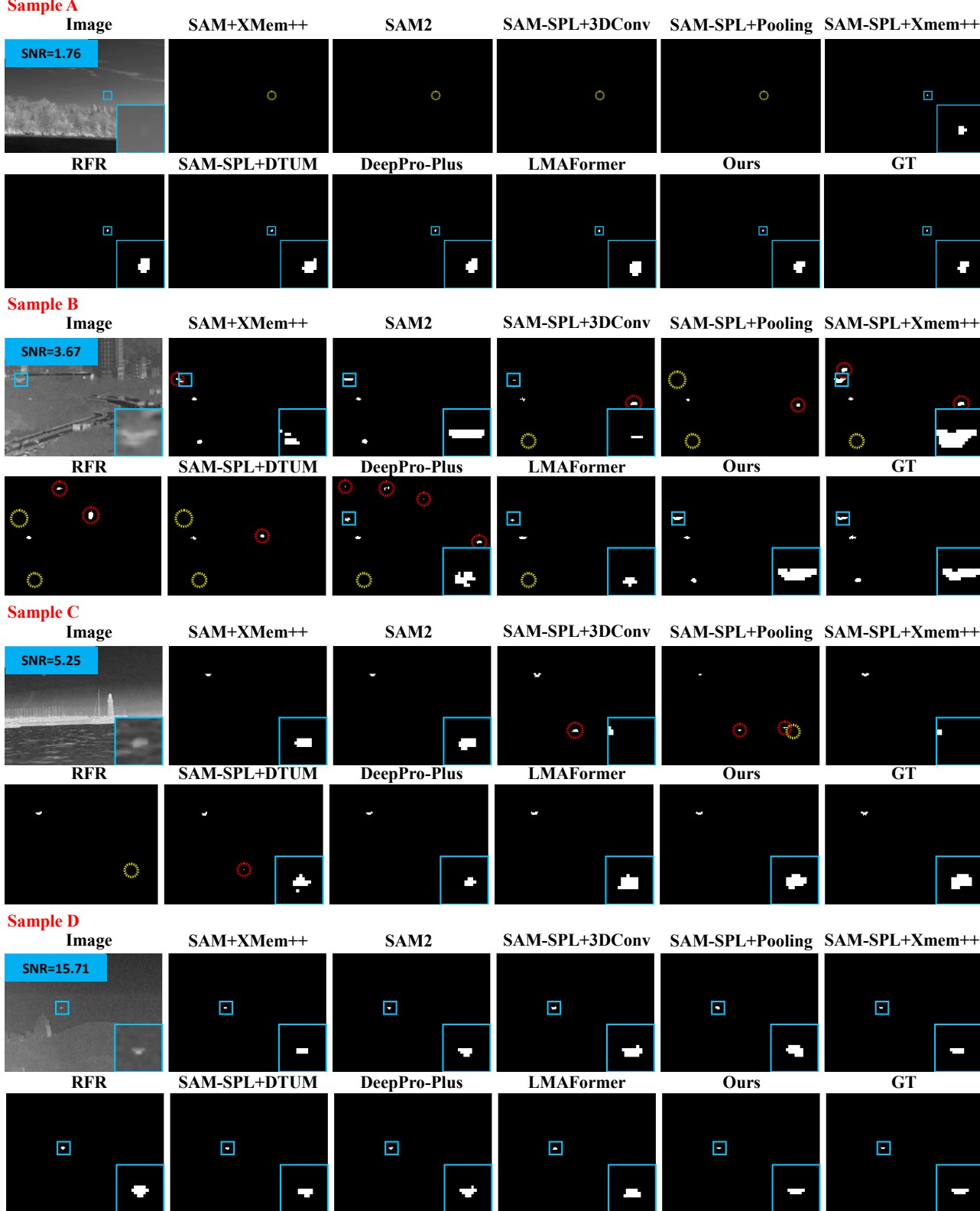

*Figure 7.* More qualitative comparisons across different SNR levels, where the red and yellow dashed circles denote the false alarms and the missed detections, respectively. One can zoom in for more details.

