# OpenReview forum: "Temporal-Emerged Prompting for Segment Anything in Multiframe Infrared Small Target Detection"
_ICML.cc/2026/Conference — ICML 2026 regular_

### Official Review · Reviewer_BgfA · 2026-03-04

**Soundness:** 2
**Presentation:** 3
**Significance:** 3
**Originality:** 2
**Overall Recommendation:** 3
**Confidence:** 3

**Summary:**

The paper addresses the important problem of object detection in infrared drone-captured imagery under adverse weather conditions. The authors propose Temporal-Emerged Prompting for Segment Anything Model (TEPSAM), a principled framework designed to explicitly exploit such temporal-emerged cues to modulate and prompt SAM. TEP-SAM operates by jointly modeling global motion patterns and local motion deviations to locate potential targets. It further enhances target region features by leveraging motion discrepancy, thereby generating temporal-emerged cues for SAM and enabling non-interactive segmentation. By bridging large-scale semantic pretraining with tasks pecific temporal modeling, TEP-SAM effectively adapts SAM to the challenging multiframe infrared small target detection task.

**Compliance With Llm Reviewing Policy:**

Affirmed.

**Ethical Review Flag:**

Flag this paper for an ethics review.

**Ethics Expertise Needed:**

["Discrimination / Bias / Fairness Concerns", "Other Expertise"]

**Final Justification:**

The paper has limited novelty and lacks proper evidence, which is given by the authors, and is not justifiable, as I am not convince their answer. So I am keeping my score the same.

**Key Questions For Authors:**

How does your model adapt to extreme intensity variations of small infrared objects under different weather and distance conditions? Is the contrast modulation dynamically adjusted or fixed?

1. Given that you adopt the Segment Anything Model pretrained on RGB natural images, how do you handle the domain gap between RGB texture-based features and texture-sparse infrared imagery?
2. What mechanisms ensure temporal consistency across frames? How do you prevent mask flickering in consecutive frames?
3. How does the framework handle long-term target disappearance and reappearance in video sequences?
4. In multi-object scenarios, how do you avoid identity switching when similar infrared targets move close to each other?
5. How is prompt drift addressed for fast-moving, pixel-scale targets? Does your method support sub-pixel prompt updating?
6. Considering the integration of SAM and temporal modules, have you evaluated the model’s computational efficiency on edge devices such as drone-based infrared platforms?

**Limitations:**

The paper is well written and explain but it has little novelty. Apart from that, some of the concerns are as follows.

1. The paper uses DETE and TFI, which rely on fixed contrast-related parameters, such as small target intensity, which varies greatly with weather and distance in real infrared scenes. A static setting may suppress extremely dim targets whose gradients fall below a threshold, and the paper does not explain how this is adaptively handled.
2. The framework uses the Segment Anything Model, which is pretrained on RGB natural images, while infrared images are texture-sparse and radiation-based, creating a domain gap. The manuscript does not clearly describe how feature mismatch or redundant representations are mitigated.
3. The method does not sufficiently address temporal consistency. Issues such as mask flickering between frames, loss of long-term memory when objects disappear and reappear, and identity switching in multi-object scenes remain unexplored.
4. The paper fails to account for prompt drift in fast-moving, pixel-scale infrared objects, as the temporal decoder lacks a sub-pixel prompt updating process, which causes the prompt to fall outside tiny object regions. Moreover, combining SAM with temporal modules increases computational cost, which conflicts with the lightweight requirements of drone-based infrared systems.

**Strengths And Weaknesses:**

Strengths:-
The paper explored good problems for object detection in adverse weather for drone imagery, which is practically relevant and under-explored compared to standard conditions. The authors systematically evaluate qualitatively and quantitatively, and perform an ablation study to evaluate each component, providing insight into what contributes to performance.

Weaknesses:-
The paper is well written and explain but it has little novelty. Apart from that, some of the concerns are as follows.

1. The paper uses DETE and TFI, which rely on fixed contrast-related parameters, such as small target intensity, which varies greatly with weather and distance in real infrared scenes. A static setting may suppress extremely dim targets whose gradients fall below a threshold, and the paper does not explain how this is adaptively handled.
2. The framework uses the Segment Anything Model, which is pretrained on RGB natural images, while infrared images are texture-sparse and radiation-based, creating a domain gap. The manuscript does not clearly describe how feature mismatch or redundant representations are mitigated.
3. The method does not sufficiently address temporal consistency. Issues such as mask flickering between frames, loss of long-term memory when objects disappear and reappear, and identity switching in multi-object scenes remain unexplored.
4. The paper fails to account for prompt drift in fast-moving, pixel-scale infrared objects, as the temporal decoder lacks a sub-pixel prompt updating process, which causes the prompt to fall outside tiny object regions. Moreover, combining SAM with temporal modules increases computational cost, which conflicts with the lightweight requirements of drone-based infrared systems.

---

> ### Author Rebuttal · Authors · 2026-03-29
>
> We sincerely thank the reviewer for the thoughtful comments.
>
> Before addressing them in detail, we‘d like to clarify three possible misunderstandings.
>
> **(1)** Our method does not rely on fixed contrast thresholds or handcrafted intensity rules; the temporal modules are **fully learnable** and optimized end-to-end from data.
>
> **(2)** TEP-SAM is not an explicit prompt-propagation or identity-tracking framework, it is designed for a detection-oriented task. It uses temporal cues to generate **internal, non-interactive** prompts for SAM to predict the target position (mask) in each frame, rather than associating object identities or propagating external prompts across frames.
>
> **(3)** The prompt in our work is a series of high-dimensional tokens derived from temporal-emerged cues, not a visible point or box prompt like in many SAM-based works.
>
> **1. Novelty and Prompting Paradigm**
>
> Regarding novelty, our method differs fundamentally from first-frame-prompted segmentation/tracking methods. For dim infrared small targets, the main difficulty is often that the target is not reliably separable from the background, sometimes even for human observers.
>
> Our contribution is not merely adding temporal modeling to SAM, but introducing a non-interactive self-prompting paradigm: consecutive frames are used to extract temporal-emerged cues, which are encoded as internal high-dimensional prompt tokens to guide SAM. Thus, our framework addresses how to generate a reliable guidance from temporal evidence itself, rather than how to propagate an already available prompt.
>
> Thus, our method does not rely on explicit coordinate-based prompts, the key issue here is not sub-pixel updating of prompt coordinates. Instead, the prompt is implicitly generated in feature space through temporally aggregated token representations, which is exactly why our framework is better suited to weak, pixel-scale infrared targets.
>
> **2. On static setting**
>
> We would like to clarify that TEP-SAM does not use fixed contrast thresholds or handcrafted intensity-related parameters. The temporal modules are fully learnable and optimized end-to-end.
>
> Mechanistically, our method relies on temporal emergence across frames instead of absolute intensity in a single frame: when a target appears brighter in some frames and dimmer in others, the temporal encoder aggregates these complementary observations and enhances weak but temporally consistent target evidence, while suppressing transient background responses. TFI then injects these aggregated cues into SAM, making the modulation implicitly data-adaptive rather than fixed.
>
> This is also consistent with Fig. 5, where our method remains effective even in low-SNR cases.
>
> **3. Domain variations**
>
> We agree that a domain gap may still exist between RGB natural images and texture-sparse infrared imagery. Therefore, we do not use SAM in an off-the-shelf manner; instead, we adapt it with temporal-emerged cues extracted from infrared sequences in TFI. In this way, TEP-SAM effectively leverages SAM’s general segmentation prior while adapting it to infrared-specific temporal evidence.
>
> **4. Temporal consistency**
>
> We would like to clarify that the core task in this paper is multiframe infrared small target **detection** rather than multi-object tracking. Our goal is to correctly segment foreground targets in each frame. This is also why the evaluation protocol focuses on detection-oriented metrics such as IoU, Pd and Fa, on which our method achieves strong performance.
>
> Therefore, ID switching caused by target disappearance/reappearance or the presence of multiple targets is considered as a **downstream tracking** problem, rather than the primary focus of this work. In this sense, our method is intended to provide strong per-frame detection/segmentation results that can serve as reliable inputs for subsequent frame-to-frame association or MOT methods.
>
> **5. Computational efficiency**
>
> We reported the parameter count, FLOPs, and FPS of TEP-SAM in the appendix, and we further provide them explicitly here for clarity.
> |Method|Params(M)|GFLOPS|FPS|
> |-|-|-|-|
> |UIUNet|50.54|436.01|21.63|
> |DNANet|4.70|114.26|12.88|
> |HCFNet|15.29|186.23|14.69|
> |SAM-SPL|12.25|30.47|32.54|
> |DeepPro-Plus|0.28|20.5|185.22|
> |DNANet+DTUM|1.21|82.80|22.09|
> |LMAFormer|390.05|380.1|4.62|
> |Ours(SAM1-Base)|5.90/95.6|876.6|12.42|
> |Ours(SAM2-Tiny)|5.88/33.10|301.1|30.23|
>
> From the model design perspective, TEP-SAM introduces only about **6M additional parameters**, so the extra overhead is relatively controlled. Moreover, the framework is compatible with lighter SAM backbones, such as smaller SAM2 variants like tiny (38.9M) and small (46M), which can further reduce the overall cost. For stricter deployment requirements, the framework can also be combined with distillation SAM backbones such as MobileSAM (9.66M). Therefore, it still has enough room for optimization toward edge deployment such as drone-based platforms.

---

> > ### Author Rebuttal · Reviewer_BgfA · 2026-04-02
> >
> > 1. How does the model behave when temporal evidence is unreliable or misleading (e.g., flickering targets, sensor noise, or abrupt motion)? Is there any mechanism to prevent temporal aggregation from amplifying noise instead of signal?
> > 2. Given that the prompts are internally generated and not user-interactive, what is the specific advantage of using SAM over a task-specific segmentation network trained directly on infrared data? Is the performance gain attributable to SAM’s prior, or to the temporal modules themselves?
> > 3. Is there any explicit mechanism (e.g., feature alignment, normalization, or adaptation loss) that ensures compatibility between SAM features and infrared temporal features? Without such alignment, how is feature mismatch avoided?
> > 4. Can the authors analyze failure cases where targets remain consistently weak across frames (i.e., no “temporal emergence”)? In such cases, does the model degrade to background noise, and how does it differ from single-frame baselines?
> > 5. How does the proposed temporal encoder compare against simpler baselines such as temporal averaging, frame differencing, or lightweight recurrent units? Can the improvement be attributed to model complexity rather than a fundamentally different mechanism?

---

> > > ### Author Response · Authors · 2026-04-03
> > >
> > > We thank the reviewer for these further questions. We respond point by point below.
> > >
> > > **Q1.**
> > > Our method does not perform unconditional temporal aggregation or naive temporal averaging, so unreliable temporal evidence is not automatically accumulated.
> > >
> > > We explicitly discourage the amplification of misleading temporal responses from three aspects.
> > >
> > > 1. DETE jointly models global temporal tendency and local temporal discrepancy to capture temporally consistent target evidence, rather than treating all temporal variations as useful cues.
> > > 2. The TDM branch explicitly captures frame-to-frame inconsistency, since misleading temporal evidence typically manifests itself as inconsistent temporal patterns.
> > > 3. The auxiliary supervision on DETE further constrains the temporal branch to learn features that are discriminative for target detection, rather than simply responding to any temporal variation.
> > >
> > > The examples in https://anonymous.4open.science/r/anonymous-6E4F (e.g., Example 3) illustrate exactly such challenging cases. As can be seen, our method remains robust under these conditions and does not exhibit the failure mode of blindly amplifying unreliable temporal evidence.
> > >
> > > **Q2.**
> > > The advantage of using SAM lies in its strong semantic segmentation prior. Compared with a task-specific segmentation network trained only on limited infrared data, this prior makes it more feasible to obtain accurate target boundaries under low-data conditions. However, the performance gain here should not be attributed to SAM alone.
> > >
> > > In M-IRSTD, targets are typically extremely weak and small, making it difficult for vanilla SAM to localize them reliably without additional task-specific guidance. As a result, SAM cannot fully leverage its segmentation prior in this setting on its own. That's why the temporal modules are necessary. Therefore, the performance gain comes from the combination of a strong segmentation prior and task-specific temporal modeling. This is already reflected in the experiments: Section 4.3 compares multiple SAM-based variants, and the ablation study in Section 4.4 further separates the contribution of each component. These results indicate that the gain arises from the interaction between SAM’s prior and the temporal modules, rather than from either part alone.
> > >
> > >
> > > **Q3.**
> > > The compatibility is enforced through both the way temporal cues are injected and the auxiliary supervision on the temporal branch. The temporal features are injected as a modulation into the frozen SAM encoder through TFI, which structurally reduces the risk of feature mismatch. Moreover, the auxiliary supervision loss on DETE requires the temporal branch to produce an auxiliary mask, directly constraining the temporal representations to be target-relevant and consistent with the segmentation objective.
> > >
> > >
> > > **Q4.**
> > > It should be clarified that our method does not rely only on obvious temporal emergence. Subtle target-related cues across frames can also be informative temporal cues. Therefore, as long as the target exhibits even slight motion or subtle temporal variation, the model can still benefit from temporal evidence, which is also demonstrated in https://anonymous.4open.science/r/anonymous-6E4F (e.g., Example 1 and 3).
> > >
> > > In the extreme case, where the targets remain completely STATIC and consistently weak across all frames, the temporal information would be invalid.  In such a case, the gap between our method and single-frame baselines may become smaller, since the prediction must rely much more on spatial evidence. However, this does not mean that the model simply degrades to background noise. Even in the absence of temporal emergence, SAM still provides a strong segmentation prior, which can exploit spatial context and boundary structure to capture the target as much as possible. We have already stated this point in the Appendix.
> > >
> > > **Q5.**
> > > We do not believe the performance gains come merely from increased model complexity. Replacing our temporal encoder with temporal average pooling causes IoU/nIoU to decrease by 21.31/15.42, while Fa increases from 0.51 to 46.00 (see the row for "w/o DETE" in Table 2 of the main text), which strongly suggests that the improvement does not arise from generic temporal accumulation.
> > >
> > > Our method explicitly combines global temporal modeling, local temporal modeling, and discrepancy-based modulation, rather than simply aggregating multi-frame features. The ablation results (Table 3 in the main text) further show that these components are complementary. Therefore, the gain comes from the specific way temporal evidence is modeled and exploited, rather than from extra model complexity.
> > >
> > > We also list the above-mentioned results in the following:
> > >
> > > |Config|IoU|nIoU|Pd|Fa|
> > > |-|-|-|-|-|
> > > |Temporal Average|64.84|70.86|96.12|46.00|
> > > |Local|80.49|82.82|98.15|9.08|
> > > |Local+Global|85.42|85.75|99.77|0.96|
> > > |Local+Global+TDM(TEP-SAM)|86.15|86.28|99.71|0.51|

---

### Official Review · Reviewer_E8WD · 2026-03-04

**Soundness:** 3
**Presentation:** 4
**Significance:** 2
**Originality:** 2
**Overall Recommendation:** 5
**Confidence:** 3

**Summary:**

The paper proposes TEP-SAM, a temporal prompting framework that injects “temporal emergence” cues into SAM for multi-frame infrared small target detection. The method builds a temporal encoder to model global background motion and local motion discrepancies, uses these features to modulate SAM’s encoder, and then generates temporal prompts to guide non-interactive segmentation. Experiments on two M-IRSTD benchmarks show clear gains over prior IRSTD methods and some SAM-based baselines, with ablations for each component.

Overall I feel the work is technically solid and quite thorough on the application side. The main concerns are around the exact assumptions behind the “temporal emergence” cue (especially under camera motion) and how general the proposed mechanism is beyond the evaluated scenarios.

**Compliance With Llm Reviewing Policy:**

Affirmed.

**Final Justification:**

The paper is technically solid, with a clear motivation, reasonable design, and experimental results that support the main claims. While the originality mainly lies in integrating existing components, it still provides a meaningful contribution to multi-frame IR small target detection, especially for adapting SAM to low-SNR scenarios.

My initial concerns about motion assumptions, applicability, and positioning relative to video-SAM were clearly addressed in the rebuttal. The clarifications and added discussion of limitations improve the paper’s clarity and positioning.

Although the scope is somewhat specialized, the solid methodology and constructive rebuttal responses make the work meet the bar for acceptance.

Therefore, my final recommendation is Accept.

**Key Questions For Authors:**

1. Assumption on motion / application scope
Your temporal emergence cue seems to rely on relatively coherent background motion plus local deviations for small targets. Does this implicitly assume a (mostly) static camera or simple background motion? What happens if the camera and small target are both moving with complex ego-motion / parallax? Please clarify the intended application scenarios and the main assumptions on camera/scene motion.

2. Relation to video-SAM / generic video segmentation
How does your design conceptually differ from simply using a video-SAM or generic video segmentation model adapted to IR data? Even a short discussion (why your temporal discrepancy + prompt pipeline is preferable in this setting) would help positioning.

**Limitations:**

The authors have not explicitly discussed the limitations or potential negative societal impact of their work. I suggest adding a short subsection that:
Clearly states the main technical limitations, e.g., that the method assumes relatively coherent background motion and may degrade under strong ego-motion or very complex scene dynamics.

**Strengths And Weaknesses:**

Soundness:
- The overall methodology is reasonable and technically sound. The temporal encoder design (global–local motion modeling + discrepancy branch) is consistent with the physical intuition that small targets deviate from background motion.
- The experimental results match the main claims: TEP-SAM consistently improves detection/segmentation metrics across benchmarks, and the ablations suggest each module (DETE, TFI, TP-Gen) contributes.
- The methods used (temporal attention / discrepancy modeling / prompt injection) are appropriate for the problem; I didn’t spot obvious technical flaws.
- There is no strong theoretical result in the strict ML sense, but for an applied vision paper the level of analysis is acceptable. The assumptions behind the temporal modeling could be spelled out more explicitly.

Presentation:
- The paper is generally clearly written and well-structured. The motivation (low-SNR small targets emerging over time) is easy to follow and the pipeline is described in a step-by-step way.
- The positioning relative to standard SAM, IRSTD baselines, and temporal methods is mostly clear, though I would appreciate a sharper contrast to video-SAM or video segmentation style methods.
- Figures and diagrams help understand how DETE/TFI/TP-Gen interact. From the description, there seems to be enough detail to re-implement the method.
- Some parts of the “temporal emergence” concept feel a bit oversold for what is, in practice, a particular design of temporal discrepancy encoder. Slightly toning down the language or clarifying the scope might help.

Significance:
- The problem (multi-frame IR small target detection under very low SNR) is important in practice and under-explored in the foundation-model era. Showing how to adapt SAM to this setting is meaningful for that community.
- The work clearly advances the state of practice for M-IRSTD and may influence follow-up work on “SAM + temporal modeling” for other small-target or low-SNR tasks.
- From a broader ML perspective, the impact is more modest: the core ideas (temporal encoder + prompt generation) are in line with existing temporal / prompting literature, and the experiments are restricted to a narrow domain.

Originality:
- The combination of SAM with a tailored “temporal emergence” encoder for IR small targets is novel in this specific domain.
- However, the building blocks (temporal attention / GRU-style aggregation / discrepancy branch / prompt tokens) are fairly standard. The originality is more in the integration and the particular way temporal cues are turned into prompts rather than in a new learning principle.
- The paper distinguishes itself from conventional IRSTD networks and from naive SAM adaptations, but the relation to video-SAM / video segmentation / temporal prompting work could be clarified more.

---

> ### Author Rebuttal · Authors · 2026-03-30
>
> We sincerely thank the reviewer for the positive assessment of our work and for these thoughtful questions. We are encouraged that the reviewer finds the method technically solid and the presentation clear. Below we clarify the main points and will revise the paper accordingly to make the scope and positioning more explicit.
>
> **1. Assumption on motion / application scope**
>
> Our method does not assume a perfectly static camera or a fully static background. A more accurate description is that TEP-SAM is designed for scenarios with normal, mild background motion within a short temporal window, such as platform jitter, slight camera motion, or smoothly varying background dynamics. From the design perspective, this is exactly why we model temporal cues using both global motion tendency and local motion discrepancy. Shared background motion can be absorbed as common temporal context, while locally inconsistent motion is highlighted as potential target evidence. In other words, the method is intended to tolerate normal slight background motion, rather than relying on an entirely motion-free scene.
>
> This scope is also consistent with the practical evaluation setting. In particular, the TSIRMT benchmark was constructed to include diverse target and background motion states, including both moving and static scenarios, rather than only perfectly static backgrounds.The background motion is modeled as random, slow, and smooth platform motion rather than strict stillness, which is also aligned with the operating regime we target. We will also provide an anonymous supplementary visualization link here to visualize successful cases under mild background motion:
> https://anonymous.4open.science/r/anonymous-6E4F
>
> At the same time, we fully agree that the assumption becomes weaker under strong ego-motion, severe parallax, or highly non-rigid background dynamics. As future work, we plan to further extend the framework to more challenging motion conditions by exploring stronger motion modeling and/or lightweight pre-alignment / registration modules before temporal self-prompting.
>
> **2.  Relation to video-SAM / generic video segmentation**
>
> Our design goal is different from generic video-SAM style methods in an important way. Most generic SAM/video-SAM pipelines are fundamentally interactive: they rely on external prompts such as points, boxes, or masks to indicate the target of interest. This setting is natural for generic object/video segmentation, but it is less suitable for M-IRSTD, where the key difficulty is precisely that the target is often too weak to be reliably discovered or manually prompted in a single frame. In contrast, our method focuses on non-interactive self-prompting: instead of assuming a visible object and then propagating a user/object prompt, we explicitly mine temporal-emerged cues from multiple frames and convert them into internal prompts for SAM. In this sense, our emphasis is not merely on adapting a generic video segmentation model to infrared data, but on addressing the more task-specific question of how to generate useful prompts when the target is initially hard to perceive.
>
> This is also where we view the main contribution of TEP-SAM: not in inventing a completely new video segmentation paradigm, but in introducing a temporal self-prompting mechanism tailored for detecting extremely weak infrared targets, where temporal emergence is used to make SAM applicable in a non-interactive M-IRSTD setting.
>
>
> **3. Limitation discussion**
>
> Thanks for the suggestion, we will add a short limitations discussion to explicitly state that the current method is best suited to scenarios with mild and relatively coherent background motion in short temporal windows, and that performance may degrade under stronger ego-motion or more complex scene dynamics. We appreciate this suggestion and believe it will improve the paper‘s clarity and scope.

---

> > ### Author Rebuttal · Reviewer_E8WD · 2026-04-02
> >
> > Thank you to the authors for the clear and well-structured rebuttal. The responses help clarify the scope and positioning of the work.
> >
> > Regarding my main questions:
> >
> > (1) Motion assumptions / application scope.
> > The authors clarified that the method is designed for short-term temporal windows with relatively coherent background motion, rather than strictly static scenes. This is consistent with the global–local temporal modeling design. The acknowledgement that performance may degrade under strong ego-motion or complex dynamics is also appropriate, and it would be good to make this explicit in the final version.
> >
> > (2) Relation to video-SAM / temporal segmentation.
> > The distinction between prompt propagation (video-SAM) and prompt generation (their temporal self-prompting mechanism) makes the contribution clearer. This helps explain why the approach is suitable for low-SNR infrared targets where reliable prompts are not available. This contrast should be made more explicit in the paper.
> >
> > (3) Limitations.
> > Adding a dedicated limitations section discussing motion assumptions and potential failure cases is important and should improve clarity.
> >
> > Overall, the rebuttal addresses my main concerns. While the scope remains somewhat specialized, the work is technically solid and well executed for the target application.
> >
> > Given that the authors have basically addressed my concerns, I can raise my score from Weak Accept to Accept.

---

> > > ### Author Response · Authors · 2026-04-03
> > >
> > > Thanks for the reviewer's willingness to raise the score from Weak Accept to Accept. Thanks again for the constructive suggestions.

---

### Official Review · Reviewer_bNg8 · 2026-03-11

**Soundness:** 2
**Presentation:** 3
**Significance:** 2
**Originality:** 2
**Overall Recommendation:** 2
**Confidence:** 4

**Summary:**

This paper addresses the multi-frame infrared small target detection (M-IRSTD) task and proposes the TEP-SAM framework.

To tackle the difficulty of distinguishing small targets from the background in a single frame, the method explicitly models Temporal-Emerged Cues to assist the foundation model (SAM) in non-interactive segmentation.

The overall system consists of three core modules:
+ DETE (Difference-Enhanced Temporal Encoder): It combines ConvGRU for global temporal modeling and extracts local motion-difference features through cosine similarity.
+ TFI (Temporal Feature Injection): It uses an MLP to inject the extracted temporal features into the intermediate layers of the frozen SAM encoder, enabling the fusion of spatial and temporal features.
+ TP-Gen (Temporal Prompt Generator): It converts temporal features into prompt tokens, replacing traditional manual clicks or bounding boxes to achieve fully automatic prediction.

**Compliance With Llm Reviewing Policy:**

Affirmed.

**Final Justification:**

Considering the limited novelty of the work, which in my view is primarily an integration of existing modules, together with its relatively narrow scope of applicability, I maintain my current rating.

**Key Questions For Authors:**

see weakness

**Limitations:**

Yes.
Although this work may have broad potential application scenarios, considering the model design and deployment cost, it remains far from practical real-world adoption. Therefore, its contribution is mainly in advancing the field of machine learning, and it does not present additional broader impact.

**Strengths And Weaknesses:**

**Strengths**:

The figures are clear and effectively illustrate both the differences between TEP-SAM and previous paradigms, as well as the internal architecture of TEP-SAM itself.

The paper is well written. The method’s structure and execution process are described clearly and precisely through extensive mathematical formulations.

The ablation study is comprehensive. By conducting extensive ablations on different modules, the paper convincingly demonstrates the effectiveness of the proposed method.

---

**Weaknesses**:

**Lack of novelty**:
Using temporal information to complement spatial information is a very common and intuitive idea [1,2,3,4], and has been widely applied across many fields. Furthermore, adding temporal modules to models such as SAM is also a common practice [5].

**Outdated temporal modeling module**:
The DETE module still relies on ConvGRU cells to process global information. However, the local receptive field of ConvGRU inherently limits its ability to capture global spatial correlations, making it naturally suboptimal as a “global” modeling module. Why not adopt more modern architectures, such as Attention or Mamba?

**Lack of practical application considerations**:
Multi-frame infrared small target detection (M-IRSTD) is likely intended for edge-device deployment scenarios. This paper introduces three additional modules to enhance SAM, which will inevitably bring considerable computational overhead and make real-time performance difficult to guarantee. The authors are encouraged to provide the parameter count, FLOPs, and inference speed of TEP-SAM.

**Citation error**:
There is a question mark in one of the citations in the right column of Lines 96–97.

[1] Zheng, Yaozong, et al. "Odtrack: Online dense temporal token learning for visual tracking." Proceedings of the AAAI conference on artificial intelligence. Vol. 38. No. 7. 2024.
[2] Xie, Jinxia, et al. "Autoregressive queries for adaptive tracking with spatio-temporal transformers." Proceedings of the IEEE/CVF conference on computer vision and pattern recognition. 2024.
[3] Kang, Ben, et al. "Exploring enhanced contextual information for video-level object tracking." Proceedings of the AAAI conference on artificial intelligence. Vol. 39. No. 4. 2025.
[4] Feng, Xiaokun, et al. "Cstrack: Enhancing rgb-x tracking via compact spatiotemporal features." arXiv preprint arXiv:2505.19434 (2025).
[5] Yang, Cheng-Yen, et al. "Samurai: Adapting segment anything model for zero-shot visual tracking with motion-aware memory." arXiv preprint arXiv:2411.11922 (2024).

---

> ### Author Rebuttal · Authors · 2026-03-29
>
> We thank the reviewer for the careful reading and comments.
>
> **1. Novelty**
>
> We agree that using temporal information to complement spatial information is common in video tasks. However, we argue that **novelty should not be judged only at this abstract level**, but together with the task setting including inputs&outputs, prompt source, and the specific role of temporal information in the model.
>
> [1-5] you listed do use temporal information, but their task definitions and input assumptions differ greatly from ours. These tracking-style paradigms mainly address how to stably match the same target in later frames when the initial position of target is given at first.
>
> While in M-IRSTD, targets are often weak, blended into the background, and hard to separate. Thus, the key challenge is not simply to “use temporal information to complement spatial information” but to convert subtle temporal discrepancy into effective internal prompts for SAM, enabling target discovery and segmentation without manual locations or templates. Our novelty mainly lies in this problem formulation and integration manner, rather than merely “adding a temporal module to SAM” **for tracking task like in [5]**. As R.E8WD noted, “The originality is more in the integration and the particular way temporal cues are turned into prompts.”
>
> To further demonstrate the inherent difference between tracking and M-IRSTD, we additionally transferred representative tracking-style methods into M-IRSTD. The results show that even when the initial target location or template is given in the first frame, directly applying tracking-style temporal modeling still cannot replace our design tailored for weak-target detection.
> |Method|IoU|nIoU|Pd|Fa↓|
> |-|-|-|-|-|
> |TEP-SAM(Ours)|86.15|86.28|99.71|0.51|
> |ODTrack|-|-|27.01|>999|
> |AQATrack|-|-|61.48|>999|
> |MCITrack|-|-|59.11|955.30|
> |SAMURAI(zero-shot)|38.80|19.89|60.44|755.53|
>
> **2. Why ConvGRU**
>
> For global temporal feature modeling, global in our design mainly refers to cross-frame accumulation along time. What we care about is long-range accumulation across frames, rather than spatial correlations over the whole image, especially given the extreme spatial sparsity of the targets.
>
> For this purpose, ConvGRU is well matched to the task. Its recurrent state and gated updates can accumulate stable temporal patterns progressively and efficiently. In contrast, Attention is stronger at relation interaction than recurrent state accumulation, and using it here tends to emphasize global pairwise matching rather than temporal memory itself, while also introducing higher computation and more redundant background interaction. Mamba often benefits more from longer 1D sequences; under our current setting with a temporal window, sparse targets, and an emphasis on stable temporal accumulation, it is not necessarily more suitable than a lightweight and explicit ConvGRU. So although these architectures are newer, they are not necessarily better matched to our model.
> |GlobalMotion|IoU|nIoU|Pd|Fa↓|
> |-|-|-|-|-|
> |ConvGRU(Ours)|86.15|86.28|99.71|0.51|
> |Attention|84.32|85.38|99.77|2.85|
> |MambaStyle|84.24|85.06|98.73|2.00|
>
> Meanwhile, our method does not exclude attention. DETE already includes patch temporal attention, and the SAM encoder/decoder is also attention-based. In other words, we use a complementary design: attention for relation modeling, and ConvGRU for temporal state accumulation. **We argue that our purpose in this paper is to how to mine the accumulated subtle temporal discrepancy and inject them into SAM as an effective internal prompts, instead of proposing a novel temporal modeling module.**
>
> **3. Computational cost**
>
> The practical scenarios of IRSTD are broad. In applications such as ground-/ship-borne infrared early warning and surveillance systems, the edge devices typically can hold one or two embedded GPUs or NPUs. For such scenarios, our method can be totally adopted.
>
> We reported the computational metrics of TEP-SAM in the appendix, and further provide trainable and total parameter count, FLOPs, and FPS explicitly here for clarity.
> |Method|Params(M)|GFLOPS|FPS|
> |-|-|-|-|
> |UIUNet|50.54|436.01|21.63|
> |DNANet|4.70|114.26|12.88|
> |HCFNet|15.29|186.23|14.69|
> |SAM-SPL|12.25|30.47|32.54|
> |DeepPro-Plus|0.28|20.5|185.22|
> |DNANet+DTUM|1.21|82.80|22.09|
> |LMAFormer|390.05|380.1|4.62|
> |Ours(SAM1-Base)|5.90/95.60|876.6|12.42|
> |Ours(SAM2-Tiny)|5.88/33.10|301.1|30.23|
>
> Moreover, TEP-SAM introduces only about 6M additional parameters, so the extra overhead is relatively controlled. Moreover, the framework is compatible with lighter SAM backbones, such as smaller SAM2 variants like tiny (38.9M) and small (46M), which can further reduce the overall cost. For stricter deployment requirements, the framework can also be combined with distillation SAM backbones such as MobileSAM (9.66M). Therefore, it still has enough room for optimization toward edge deployment.
>
> **4. Citation error**
>
> We will correct the typo in the revision.

---

> > ### Author Rebuttal · Reviewer_bNg8 · 2026-04-03
> >
> > Thank you for your response. However, I still find that my concerns have not been fully addressed.
> >
> > First, the claimed novelty of the paper remains limited in my view. The current method appears to rely primarily on stacking multiple existing modules, rather than introducing a genuinely new technical perspective or a substantially original methodological insight. While such integration may bring empirical improvements, it is difficult to see how this design can provide broader inspiration to the community beyond task-specific engineering.
> >
> > Second, the practical applicability of the proposed method is still unclear. In particular, the reported TEP-SAM variants based on the Base and Large architectures seem difficult to meet the real-time requirements that are often crucial in infrared small target detection scenarios, which leaves the practical value of the method still quite limited.

---

> > > ### Author Response · Authors · 2026-04-03
> > >
> > > **1. About the Novelty.**
> > >
> > > We have already clarified this distinction in the rebuttal and further strengthened it by **adding direct comparisons with representative tracking-style methods raised by the reviewer.** Those additional results show that even when tracking methods are given favorable assumptions, such as an initial target location or template in the first frame, they still do not match the proposed design in the M-IRSTD setting. This suggests that the contribution is not merely empirical engineering, but a task-specific methodological adaptation that addresses a genuinely different problem. Moreover, the Cross-Task Visualization on low-visibility detection/segmentation problems at this link https://anonymous.4open.science/r/anonymous-6E4F also shows that our method may provide broader inspiration to the community beyond M-IRSTD task.
> > >
> > > We respectfully disagree with the reviewer’s assessment that our novelty is limited. The reviewer’s expectation—demanding a "genuinely new technical perspective" or a "substantially original methodological insight"—sets a bar that is higher than the conference’s official reviewer guidelines. **As explicitly stated in the reviewer guidelines: "originality does not necessarily require introducing an entirely new method. Rather, a work that provides novel insights by evaluating existing methods, or demonstrates improved efficiency, fairness, etc. is also equally valuable."** Our work falls precisely into this recognized category: we do not merely "stack existing modules" in a trivial way; we provide a non-trivial integration that yields clear empirical improvements and offers design insights beyond task-specific engineering. Moreover, the reviewer’s comment on our "limitations" section suggests that the evaluation was not conducted under the official reviewer guidelines. We therefore kindly ask the reviewer to reassess our contribution according to the conference’s own definition of originality, rather than a stricter, unpublished standard.
> > >
> > >
> > > **2. About the Practical Applicability.**
> > >
> > > We respectfully believe this concern may rely on an overly narrow assumption about the intended deployment scenario of MIRSTD. While edge deployment and real-time processing are indeed important in some applications, **MIRSTD is not exclusively defined by strict edge-device or hard real-time constraints.** In practice, infrared small target detection is also used in scenarios such as ground-/ship-borne early warning and surveillance systems, where the hardware budget is broader and detection accuracy remains the primary concern. Importantly, the field is still largely performance-driven; improving detection accuracy is currently the central focus for most published works, and real-time requirements are not the dominant evaluation criterion.
> > >
> > > Nevertheless, we have not ignored efficiency. Our Base variant already achieves 12FPS on 1–2 GPUs. With standard engineering optimizations (e.g., model pruning, TensorRT acceleration, mixed-precision inference), we are confident that the same model can reach ~30FPS without significant accuracy loss. Moreover, as noted in our paper and appendix, the additional overhead of TEP-SAM is relatively controlled (about 6M extra parameters), and the framework is compatible with lighter SAM backbones, including smaller SAM2 variants and distilled versions such as MobileSAM. This leaves clear room for deployment-oriented optimization.
> > >
> > > Therefore, we do not think it is fair to conclude that the practical value is "quite limited" simply because the current method is not yet optimized for the most restrictive real-time edge scenario. We agree that efficiency is an important direction for future improvement, but we believe the method already provides a strong accuracy-speed trade-off, and its practical value is well within the scope and claims of the paper.
> > >
> > > **We sincerely hope that the reviewer could kindly reconsider the contribution and practical value of our work. A positive reassessment and an increase in the score would be greatly appreciated. We are grateful for the reviewer’s time and constructive feedback.**

---

### Official Review · Reviewer_3s8G · 2026-03-13

**Soundness:** 3
**Presentation:** 3
**Significance:** 3
**Originality:** 3
**Overall Recommendation:** 5
**Confidence:** 5

**Summary:**

This paper proposes TEP-SAM for multi-frame infrared small target detection under extremely low SNR. To overcome single-frame clutter, TEP-SAM introduces a temporal-emerged prompting paradigm that automatically derives prompts from motion dynamics. Experiments on M-IRSTD benchmarks show that TEP-SAM significantly outperforms classical methods and SAM-based baselines.

**Compliance With Llm Reviewing Policy:**

Affirmed.

**Final Justification:**

Thank the authors for their rebuttal. I will keep my score.

**Key Questions For Authors:**

- The framework assumes that targets exhibit motion discrepancy relative to background dynamics. How does TEP-SAM perform when targets are stationary or exhibit minimal motion?
-  In the comparisons with SAM-variant, many SAM baselines rely on external prompts, whereas TEP-SAM generates internal prompts. Could stronger learned prompting baselines reduce the observed performance gap?
-  Have the authors investigated whether temporal-emerged prompting can benefit other low-visibility segmentation tasks?

**Limitations:**

yes

**Strengths And Weaknesses:**

Pros
- This temporal-emerged prompting framework provides meaningful insights into tasks with weak spatial signals.
- The proposed method is well motivated and technically sound.
- The experiments are comprehensive. Comparisons with state-of-the-art methods, including the SAM-based variants, are convincing.
- This paper is well organized and easy to follow.

Cons
- The method appears to rely on motion discrepancy between the target and the background. Its performance on stationary or slowly moving targets is not fully explored and could be further clarified.
- It would be interesting to better understand the generalization ability of the proposed method on other related tasks involving weak or small targets.

---

> ### Author Rebuttal · Authors · 2026-03-31
>
> We sincerely thank the reviewer for the positive assessment of our work and for the constructive questions. We are encouraged that the reviewer finds the temporal-emerged prompting framework meaningful, technically sound, and empirically convincing. Below, we respond to the three main questions in turn.
>
> **When targets are stationary or exhibit minimal motion**
>
> Our method is built on temporal-emerged cues, and thus it benefits most when the target exhibits distinguishable temporal discrepancy relative to the background. When the target moves extremely slowly or becomes nearly stationary, such temporal cues may become much weaker, and the effectiveness of our temporal module may correspondingly decrease. In this sense, we agree that this is a limitation of the current version.
>
> As future work, we plan to further investigate this challenging setting by exploring more adaptive temporal modeling strategies, such as dynamic sampling windows, adaptive frame intervals, and combined short-term/long-term temporal modeling, so that the model can better handle slowly moving or near-stationary targets.
>
> **Learned prompting baselines**
>
> We agree that stronger learned prompting baselines are important to examine. To better address this point, we additionally evaluate SAM-I2V under both zero-shot and fine-tuned settings, using either an initial point or box prompt on the first visible frame. The results show that stronger learned external prompting can indeed improve performance, especially after fine-tuning and with box initialization.
>
> |Method|IoU|nIoU|Pd|Fa↓|
> |-|-|-|-|-|
> |SAM-I2V(zero-shot) init point|16.52|0.06|19.55|>999|
> |SAM-I2V(zero-shot) init box|43.23|42.21|42.28|0.32|
> |SAM-I2V(fine-tuned) init point|55.21|3.19|68.42|>999|
> |SAM-I2V(fine-tuned) init box|62.41|55.79|81.26|169.69|
> |TEP-SAM(Ours)|86.15|86.28|99.71|0.51|
>
> At the same time, a clear gap to TEP-SAM still remains. We believe this further supports our motivation: in the challenging M-IRSTD setting, where targets are often difficult to perceive reliably from a single frame, deriving prompts automatically from temporal emergence is a more suitable solution than relying on externally provided initialization alone.
>
> **Other segmentation tasks**
>
> We have conducted preliminary explorations on several related low-visibility or weak-saliency segmentation settings, including salient object detection (SOD), camouflaged object detection (COD), and remote sensing object segmentation. In particular, COD and some remote sensing scenarios are closely related to our motivation, since the objects are often embedded in highly complex backgrounds and are not easy to identify directly.
>
> Our preliminary observations suggest that the core idea of temporal-emerged/self-derived prompting can also be helpful for such problems, especially when the target is visually ambiguous and requires stronger contextual or cross-frame evidence to be revealed. We provide an anonymous link with some qualitative visualization results on these related tasks here:
> https://anonymous.4open.science/r/anonymous-6E4F
>
> These results are still preliminary, but they suggest that the proposed idea is not limited to M-IRSTD only, and may be effective more broadly for similar low-visibility detection/segmentation problems. We will clarify this point and present it as a promising future direction rather than as a fully established claim in the current paper.
>
> In summary, we thank the reviewer again for these thoughtful questions. We agree that stationary/near-stationary targets and stronger prompting baselines deserve further study, and we are also encouraged by the preliminary transferability of temporal-emerged prompting to other low-visibility tasks. We will incorporate these clarifications and discussions in the revised paper.

---

> > ### Author Rebuttal · Reviewer_3s8G · 2026-04-02
> >
> > The authors fully resolved my questions. I will keep my scores

---

### Decision · Program_Chairs · 2026-04-30

**Decision:**

Accept (regular)

**Comment:**

This paper proposes a multi-frame infrared small target detection method by modeling global motion patterns and local motion deviations to locate potential targets. After rebuttal and discussion, the paper has mixed scores. The negative reviewers mainly have concerns on the technical contributions as they thought the high-level idea has been explored and the proposed modules shared similar principals to exiting techniques, as well as its relatively slow computation that hinders its application in edge-devices. On the positive side, the reviewers acknowledged that the paper is technically sound with reasonable integration of existing techniques for addressing the new problem, and the experimental designs are reasonable which support the paper’s claims and contributions. After read the paper, reviews and discussions as well as the authors’ rebuttal, the AC agreed more on the positive side that the paper makes reasonable contributions to a meaningful problem, with persuasive results to support the main claims. The rebuttal addressed most concerns about the clarity of the method. Therefore, the AC would like to recommend accepting it.